# LoRA-DA: Data-Aware Initialization for Low-Rank Adaptation via Asymptotic Analysis

Qingyue Zhang [* 1]  Chang Chu [* 1]  Tianren Peng [1]  Qi Li [1]  Xiangyang Luo [1]  Zhihao Jiang [1]  Shao-Lun Huang [1]

## Abstract

LoRA has become a widely adopted method for PEFT, and its initialization methods have attracted increasing attention. However, existing methods have notable limitations: many methods do not incorporate target-domain data, while gradient-based methods exploit data only at a shallow level by relying on one-step gradient decomposition. In this paper, we establish a theoretical framework for data-aware LoRA initialization. Starting from minimizing the expectation of the parameter discrepancy between the fine-tuned and target models, we derive an optimization problem with two components: a bias term, which is related to the parameter distance between the fine-tuned and target models, and is approximated using a Fisher–gradient formulation to preserve anisotropy; and a variance term, which accounts for the uncertainty introduced by sampling stochasticity through the Fisher information. Solving this problem yields an optimal initialization strategy for LoRA, based on which we develop an efficient algorithm, LoRA-DA. Empirical results across multiple benchmarks demonstrate that LoRA-DA consistently improves final accuracy over existing initialization methods. Additional studies show faster, more stable convergence, robustness across ranks, and only a small initialization overhead for LoRA-DA. The source code is available at https://github.com/zqy0126/LoRA-DA.

## 1. Introduction

In recent years, large language models (LLMs) have advanced at an unprecedented pace, reshaping research in natural language processing and neighboring areas. By scaling parameters, data, and compute, LLMs exhibit strong generalization across diverse domains. Representative studies report compelling progress in instruction-following dialogue (Ouyang et al., 2022), code reasoning and synthesis (Li et al., 2022; Wang et al., 2021), commonsense reasoning (Toroghi et al., 2025), mathematical problem solving (Wei et al., 2022; Wang et al., 2023a; Hendrycks et al., 2021), and multimodal understanding with vision–language modeling (Li et al., 2023; Dai et al., 2023). However, these models often require full-parameter fine-tuning, the cost of which is prohibitively large.

To alleviate the prohibitive cost of full-parameter fine-tuning, a growing body of research has focused on parameter-efficient fine-tuning (PEFT) techniques, which adapt large models by introducing only a small set of trainable parameters. Among them, Low-Rank Adaptation (LoRA) has emerged as a highly influential approach, which injects low-rank matrices into pretrained weight space to enable efficient adaptation without modifying the original parameters (Hu et al., 2022). Moreover, LoRA-FA simplifies LoRA by freezing $A$ and training only $B$, cutting trainable parameters by half while maintaining competitive performance. Despite these advantages, LoRA and LoRA-FA share a common initialization scheme: the $A$ matrix is randomly initialized (or frozen as random in LoRA-FA), while the $B$ matrix is initialized to zero. Such a scheme ensures that no adaptation is applied at the very beginning of training, but it also introduces two drawbacks (Meng et al., 2024; Zhang et al., 2025b): (i) the training process starts slowly due to the absence of informative initialization, and (ii) the models may fail to converge to an optimal solution.

Compared with conventional random initialization in LoRA, recent work has highlighted the critical role of initialization for both convergence speed and final performance, motivating alternative initialization strategies. Prior approaches were generally data-agnostic, without incorporating information from the target task, such as PiSSA (Meng et al., 2024) and MiLoRA (Wang et al., 2025), which adapt the singular vectors or singular values of pretrained weight matrices to exploit the inherent structural properties of the original parameters. More recently, several data-aware

---

[*]Equal contribution  [1]Tsinghua Shenzhen International Graduate School, Tsinghua University, Shenzhen, China. Correspondence to: Shao-Lun Huang <twn2gold@gmail.com>.

*Proceedings of the 43rd International Conference on Machine Learning*, Seoul, South Korea. PMLR 306, 2026. Copyright 2026 by the author(s).

approaches have been proposed that leverage a small set of target-domain samples for initialization. For example, LoRA-GA (Wang et al., 2024a) and LoRA-One (Zhang et al., 2025b) exploit target-domain gradients to construct the low-rank subspace. However, their own experiments show that the performance of one-step fine-tuning is not only suboptimal but also markedly inferior to that of vanilla LoRA, raising concerns about whether gradient-only initialization is sufficient. Moreover, their conclusions either lack rigorous theoretical support or rely on strong mathematical assumptions about the input vector of the layer, such as isotropic centered sub-Gaussian assumption. However, recent empirical studies have shown that representations in transformer-based models are far from isotropic (Godey et al., 2024). In our view, relying solely on gradients to approximate the parameter discrepancy overlooks the anisotropy of the parameter space. Moreover, beyond parameter discrepancy, the variance induced by sampling stochasticity also contributes to the training error, yet such methods fail to account for it in their initialization strategies.

In this work, we propose a data-aware LoRA initialization method grounded in asymptotic analysis. Specifically, we formulate the optimization objective as minimizing the expectation of the discrepancy between the parameters of fine-tuned and target models. By applying asymptotic analysis, we reformulate the optimization of the bound on this objective into a quadratic optimization problem with the LoRA initialization parameters as target variables. The **Initialization Guidance Matrix**, serving as the coefficient matrix in the quadratic optimization, consists of two terms: a **variance term**, capturing the uncertainty caused by sampling stochasticity through Fisher information, and a **bias term**, which relates to the discrepancy between the fine-tuned and target models, approximated via a Fisher–gradient approach that preserves the anisotropic structure of the parameter space. Solving this problem yields an optimal initialization strategy for LoRA. Building upon this theory, we propose LoRA-DA, a general data-aware LoRA initialization algorithm. Extensive experiments across multiple tasks demonstrate the effectiveness and robustness of our algorithm.

In summary, our contributions are as follows:

1. **Theoretical foundation.** We establish a theoretical framework for data-aware LoRA initialization based on asymptotic analysis. Our formulation decomposes the bound on expected estimation error into a variance term and a bias term, yielding a method for computing the optimal LoRA initialization.

2. **Algorithm design.** Building on these theoretical insights, we propose LoRA-DA, a practical algorithm that leverages a small set of target-domain samples to estimate the statistics required by our theoretical framework and derive the optimal initialization of LoRA.

The proposed initialization algorithm is architecture-agnostic.

3. **Empirical validation.** We conduct extensive experiments on both natural language understanding benchmarks and natural language generation benchmarks, where LoRA-DA consistently outperforms state-of-the-art initialization methods, achieving average improvements of **0.3%** on natural language understanding and **1.0%** on natural language generation over prior SOTA with the 7B model. Additional studies confirm the applicability of LoRA-DA to larger model, as well as faster and more stable convergence, robustness across different ranks, and a small initialization overhead.

**Conflict of Interest Disclosure:** The authors declare that they have no financial conflicts of interest related to this work.

## 2. Preliminaries

In this section, we introduce the mathematical preliminaries required for our theoretical framework. Section 2.1 provides the formal definition and background of LoRA and LoRA-FA. Section 2.2 presents the asymptotic normality of the maximum likelihood estimator (MLE) used to model sampling stochasticity, and the Fisher information matrix which is not only related to asymptotic normality but also used to model the anisotropy of the parameter space. Finally, Section 2.3 reviews the correspondence between the decomposition of eigenvalues and the solution of the quadratic optimization problem, which we will exploit in our method.

### 2.1. Low-Rank Adaptation (LoRA) and LoRA-FA

LoRA enables PEFT of pre-trained networks by inserting low-dimensional trainable components. Instead of fine-tuning all existing weights, LoRA appends two compact matrices $\boldsymbol{A} \in \mathbb{R}^{d_1 \times r}$ and $\boldsymbol{B} \in \mathbb{R}^{r \times d_2}$, with $r \ll \min(d_1, d_2)$. The weight adaptation of LoRA can be expressed as

$$Y = Z\hat{\boldsymbol{W}} = Z(\boldsymbol{W}_0 + \Delta\boldsymbol{W}) = Z(\boldsymbol{W}_0 + \boldsymbol{AB}), \quad (1)$$

where $Z$ is the input, $Y$ is the output, and $\Delta\boldsymbol{W}$ denotes the low-rank update. Typically, $\boldsymbol{A}$ is initialized from a Kaiming uniform distribution (He et al., 2015), and $\boldsymbol{B}$ is initialized to zeros .

LoRA-FA (LoRA with Frozen-A) (Zhang et al., 2023a) is a memory-efficient variant of standard LoRA. In conventional LoRA, both low-rank matrices $\boldsymbol{A}$ and $\boldsymbol{B}$ are trained, requiring storage for gradients and optimizer states of both. LoRA-FA freezes $\boldsymbol{A}$ after initialization, and only updates $\boldsymbol{B}$. This eliminates the need to store $\boldsymbol{A}$'s activations and optimizer states, substantially reducing fine-tuning memory usage. The design of LoRA-FA is further supported

by empirical and theoretical evidence. The original LoRA work (Hu et al., 2022) found that assigning a larger learning rate to $\boldsymbol{B}$ than to $\boldsymbol{A}$ leads to better performance, suggesting that $\boldsymbol{B}$ contributes more to adaptation. More recent work (Zhu et al., 2024) systematically studied this asymmetry, showing that training $\boldsymbol{B}$ is inherently more effective, while frozen $\boldsymbol{A}$ has little impact on final accuracy.

## 2.2. Asymptotic Normality of the Maximum Likelihood Estimator (MLE)

When estimating the underlying parameter $\boldsymbol{\theta}^* \in \mathbb{R}^d$ from independent and identically distributed (i.i.d.) samples drawn from $P_{X;\boldsymbol{\theta}^*}$, we denote $\mathcal{D}$ as a set of $N$ i.i.d. samples from the distribution. Then the MLE can be expressed as

$$\hat{\boldsymbol{\theta}}_{MLE} = \arg\max_{\boldsymbol{\theta}} \frac{1}{N} \sum_{x \in \mathcal{D}} \log P_{X;\boldsymbol{\theta}}(x). \quad (2)$$

Under standard regularity conditions, the MLE satisfies the following asymptotic normality (Van der Vaart, 2000):

$$\sqrt{N}\left(\hat{\boldsymbol{\theta}}_{\mathrm{MLE}} - \boldsymbol{\theta}^*\right) \xrightarrow{d} \mathcal{N}\left(0, \boldsymbol{J}(\boldsymbol{\theta}^*)^{-1}\right), \quad (3)$$

where "$-1$" denotes the matrix inverse and $\boldsymbol{J}(\boldsymbol{\theta})$ is the Fisher information matrix defined as:

$$\boldsymbol{J}(\boldsymbol{\theta})^{d \times d} = \mathbb{E}\left[\left(\frac{\partial}{\partial \boldsymbol{\theta}} \log P_{X;\boldsymbol{\theta}}\right)\left(\frac{\partial}{\partial \boldsymbol{\theta}} \log P_{X;\boldsymbol{\theta}}\right)^{\top}\right]. \quad (4)$$

Intuitively, the Fisher information matrix measures the sensitivity of the model to perturbations along different parameter directions, thus serving as a descriptor of the anisotropy in the parameter space.

## 2.3. Eigenvalue Decomposition for Quadratic Form Minimization

We consider the constrained quadratic minimization problem

$$\min_{\boldsymbol{Q} \in \mathbb{R}^{d \times r}} \operatorname{tr}(\boldsymbol{Q}^{\top} \boldsymbol{M} \boldsymbol{Q}) \quad \text{s.t.} \quad \boldsymbol{Q}^{\top}\boldsymbol{Q} = \boldsymbol{I}_r, \quad (5)$$

where $\boldsymbol{M} \in \mathbb{R}^{d \times d}$ is symmetric. By the eigenvalue decomposition

$$\boldsymbol{M} = \boldsymbol{U}\boldsymbol{\Lambda}\boldsymbol{U}^{\top}, \quad \boldsymbol{\Lambda} = \operatorname{diag}(\lambda_1, \ldots, \lambda_d), \quad \lambda_1 \geq \cdots \geq \lambda_d, \quad (6)$$

the Courant–Fischer min–max theorem (Horn & Johnson, 2012) ensures that the optimal solution is

$$\boldsymbol{Q}^* = \begin{bmatrix} \boldsymbol{u}_d & \boldsymbol{u}_{d-1} & \cdots & \boldsymbol{u}_{d-r+1} \end{bmatrix}, \quad (7)$$

where $\boldsymbol{u}_i$ is the eigenvector corresponding to the eigenvalue $\lambda_i$. The minimum objective value is

$$\min_{\boldsymbol{Q}} \operatorname{tr}(\boldsymbol{Q}^{\top}\boldsymbol{M}\boldsymbol{Q}) = \sum_{i=d-r+1}^{d} \lambda_i. \quad (8)$$

## 3. Problem Formulation

Let $P_{X,\boldsymbol{W}_0}$ denote a pre-trained model, and $P_{X,\boldsymbol{W}_{\mathrm{tgt}}}$ denotes the target model in the fine-tuning process, where $\boldsymbol{W} \in \mathbb{R}^{d_1 \times d_2}$ is the parameter matrix, and $X$ represents the joint distribution of inputs and outputs. For example, when the task is a supervised classification task, $P_{X,\boldsymbol{W}_0}$ corresponds to the joint distribution model of input features $Z$ and output labels $Y$, i.e., $X = (Z, Y)$. Suppose that we are given a set of $N$ samples $\{x_1, \ldots, x_N\}$ i.i.d. drawn from $P_{X,\boldsymbol{W}_{\mathrm{tgt}}}$ for the fine-tuning of LoRA, whose form is

$$\hat{\boldsymbol{W}} = \boldsymbol{W}_0 + \boldsymbol{A}\boldsymbol{B}, \quad \boldsymbol{A} \in \mathbb{R}^{d_1 \times r}, \ \boldsymbol{B} \in \mathbb{R}^{r \times d_2}, \quad (9)$$

where $r \ll \min(d_1, d_2)$. Fine-tuning with cross-entropy is equivalent to MLE, while LoRA restricts updates to a low-rank subspace, yielding a constrained MLE:

$$\hat{\boldsymbol{W}} = \arg\max_{\boldsymbol{W} \in \left\{\boldsymbol{W}_0 + \boldsymbol{A}\boldsymbol{B} \,\middle|\, \boldsymbol{A} \in \mathbb{R}^{d_1 \times r}, \boldsymbol{B} \in \mathbb{R}^{r \times d_2}\right\}} \frac{1}{N} \sum_{i=1}^{N} \log P_{X,\boldsymbol{W}}(x_i). \quad (10)$$

We first investigate the initialization of matrix $\boldsymbol{A}$ under the LoRA-FA framework, i.e., the setting where $\boldsymbol{A}$ remains frozen during fine-tuning. Considering the practical similarity between LoRA-FA and full LoRA in terms of both training behavior and empirical performance, as illustrated in Subsection 2.1, our analysis in the LoRA-FA setting does not compromise generality. The initialization strategy derived from this theoretical analysis is not restricted to LoRA-FA but can also be directly applied to standard LoRA, as validated empirically in Section 5. In addition, we constrain the $\boldsymbol{A}$ to be column-orthogonal, i.e. $\boldsymbol{A}^{\top}\boldsymbol{A} = \boldsymbol{I}_r$, which prevents redundancy among low-rank directions, and facilitates better-conditioned optimization. Our objective is to design a more effective initialization $\boldsymbol{A}_0$ such that the resulting estimator $\hat{\boldsymbol{W}}$ minimizes the expected squared Frobenius norm to the true parameter $\boldsymbol{W}_{\mathrm{tgt}}$, i.e.,

$$\boldsymbol{A}_0^* = \arg\min_{\boldsymbol{A}} \mathbb{E}\left[\left\|\hat{\boldsymbol{W}} - \boldsymbol{W}_{\mathrm{tgt}}\right\|_F^2\right], \quad (11)$$

where $\|\cdot\|_F$ denotes the Frobenius norm (Horn & Johnson, 2012), and $\hat{\boldsymbol{W}}$ is defined as (10), thereby being associated with $\boldsymbol{A}$. To facilitate the subsequent mathematical derivations, we assume that the distance between the parameters of the target sample model and the source pre-trained model is sufficiently small, i.e., $\|\boldsymbol{W}_{\mathrm{tgt}} - \boldsymbol{W}_0\|_F = O\left(\frac{1}{\sqrt{N}}\right)$. This assumption is made without loss of generality, since fine-tuning typically targets tasks near the pre-training model. Beyond the well-known similarity in early layers (Raghu et al., 2019), cross-layer evidence shows that keeping weights close to the pre-trained parameters benefits fine-tuning stability and generalization (Lee et al., 2020; Chen et al., 2020). Similar assumptions have also been adopted in the transfer learning literature (Zhang et al., 2025a).

# 4. Main Result

In this section, we present the procedure by which the optimal initialization of LoRA is derived through minimizing the objective function (11). To make the mathematical derivation and results more accessible to the reader, we first provide the theoretical analysis in the simplified case where the output dimension is $d_2 = 1$ in Section 4.1. We then extend the result to the general high-dimensional setting of standard LoRA in Section 4.2, and finally describe the practical algorithm LoRA-DA in Section 4.3.

## 4.1. One-Dimensional Case

We first consider the simple case where the output dimension is $d_2 = 1$, i.e., $\boldsymbol{W} \in \mathbb{R}^{d_1 \times 1}$. In this setting, the estimator can be expressed as $\hat{\boldsymbol{W}} = \boldsymbol{W}_0 + \boldsymbol{A}\boldsymbol{B}$, where $\boldsymbol{A} \in \mathbb{R}^{d_1 \times r}$ is a fixed column-orthogonal matrix and $\boldsymbol{B} \in \mathbb{R}^{r \times 1}$. Then we have the following theorem.

**Theorem 4.1.** *(proved in Appendix C.1) In LoRA fine-tuning from $P_{X;\boldsymbol{W}_0}$ to $P_{X;\boldsymbol{W}_{\mathrm{tgt}}}$, we assume $\boldsymbol{W}_0, \boldsymbol{W}_{\mathrm{tgt}} \in \mathbb{R}^{d_1 \times 1}$, and $\|\boldsymbol{W}_{\mathrm{tgt}} - \boldsymbol{W}_0\| = O\left(\frac{1}{\sqrt{N}}\right)$. Then, the optimal initialization of the LoRA matrix $\boldsymbol{A}$ is given by*

$$\boldsymbol{A}_0^* = \arg\min_{\boldsymbol{A}} \mathrm{tr}\left(\boldsymbol{A}^\top \boldsymbol{\Omega} \boldsymbol{A}\right), \quad (12)$$

*and from Section 2.3 we know that the $r$ column vectors of $\boldsymbol{A}_0^*$ correspond to the eigenvectors of $\boldsymbol{\Omega}$ associated with its $r$ smallest eigenvalues, where $\boldsymbol{\Omega}$ is the **Initialization Guidance Matrix** given by*

$$\boldsymbol{\Omega}^{d_1 \times d_1} = \left(\underbrace{\frac{\boldsymbol{J}(\boldsymbol{W}_0)^{-1}}{N}}_{variance\ term} \underbrace{-\left(\boldsymbol{W}_{\mathrm{tgt}} - \boldsymbol{W}_0\right)\left(\boldsymbol{W}_{\mathrm{tgt}} - \boldsymbol{W}_0\right)^\top}_{bias\ term}\right)$$
$$(13)$$

In Fig. 1, we provide an explanation of the result in Theorem 4.1. Specifically, the bound on the objective function in (11) can be decomposed into two components: a variance term models the sampling stochasticity associated with the Fisher information and sample size, and a bias term arising from the discrepancy between the target parameter $\boldsymbol{W}_{\mathrm{tgt}}$ and the LoRA subspace determined by $\boldsymbol{W}_0$ and fixed $A$. It should be noted that the bias term in (13) excludes the invariant component of the optimization, and thus appears with a minus sign. Its complete form can be found in (46).

In (13), a natural question arises: how to approximate the term $\boldsymbol{W}_{\mathrm{tgt}} - \boldsymbol{W}_0$, i.e., the discrepancy between the target parameters $\boldsymbol{W}_{\mathrm{tgt}}$ and the pre-trained parameters $\boldsymbol{W}_0$. Prior works often approximate this discrepancy directly from the negative raw gradient. We instead adopt a more effective approach, namely the **Fisher-gradient**, approximating it as the negative inverse Fisher times the gradient. Unlike the

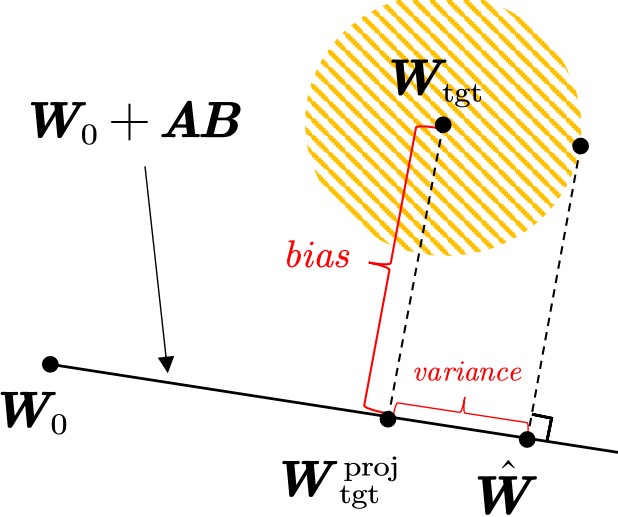

*Figure 1.* The yellow circle illustrates the estimation variance induced by the stochasticity of training samples in the unconstrained setting. The red variance term represents its projection onto the LoRA subspace under the fixed-$\boldsymbol{A}$ constraint, while the red bias term corresponds to the approximation error due to the distance between $\boldsymbol{W}_{\mathrm{tgt}}$ and the LoRA subspace.

raw gradient, this Fisher-weighted form adaptively scales directions by their uncertainty or information content, thereby capturing the model's anisotropy. The Fisher-gradient formulation builds on the classical notion of the natural gradient (Amari, 1998). In our framework, the Fisher matrix is already obtained in the process of evaluating the variance term and can therefore be reused, which provides a natural motivation for incorporating the Fisher-gradient formulation into our initialization strategy. We next provide the theoretical justification and formulation of this approach.

We employ a local second-order expansion around $\boldsymbol{W}_0$. Specifically, we define the empirical loss on the target distribution as $\mathcal{L}_{\mathrm{tgt}}(\boldsymbol{W}) = \frac{1}{N}\sum_{j=1}^{N}\ell(\boldsymbol{W}; z_{\mathrm{tgt}}^j, y_{\mathrm{tgt}}^j)$, where the subscript "tgt" indicates that the loss is computed on the fine-tuning samples drawn from $P_{X,\boldsymbol{W}_{\mathrm{tgt}}}$. Expanding this loss around $\boldsymbol{W}_0$ yields

$$\mathcal{L}_{\mathrm{tgt}}(\boldsymbol{W}_{\mathrm{tgt}}) \approx \mathcal{L}_{\mathrm{tgt}}(\boldsymbol{W}_0) + \boldsymbol{G}^\top(\boldsymbol{W}_{\mathrm{tgt}} - \boldsymbol{W}_0) +$$
$$\frac{1}{2}(\boldsymbol{W}_{\mathrm{tgt}} - \boldsymbol{W}_0)^\top \boldsymbol{H}_0(\boldsymbol{W}_{\mathrm{tgt}} - \boldsymbol{W}_0), \quad (14)$$

where $\boldsymbol{G}^{d_1 \times 1} = \nabla\mathcal{L}_{\mathrm{tgt}}(\boldsymbol{W}_0)$ is the gradient evaluated at the pre-trained parameters and $\boldsymbol{H}_0$ denotes the Hessian computed at $\boldsymbol{W}_0$. The first-order optimality condition for the target optimum $\boldsymbol{W}_{\mathrm{tgt}}$ is written as

$$\nabla_{\boldsymbol{W}}\mathcal{L}_{\mathrm{tgt}}(\boldsymbol{W}_{\mathrm{tgt}}) = 0 \approx \boldsymbol{G} + \boldsymbol{H}_0(\boldsymbol{W}_{\mathrm{tgt}} - \boldsymbol{W}_0). \quad (15)$$

Solving for the displacement gives $\boldsymbol{W}_{\mathrm{tgt}} - \boldsymbol{W}_0 \approx -\boldsymbol{H}_0^{-1}\boldsymbol{G}$. In practice, the Hessian $\boldsymbol{H}_0$ is typically approximated by the Fisher information matrix (He et al., 2024).

This is theoretically justified under standard regularity conditions for MLE with i.i.d. data, where the Fisher information coincides with the expected Hessian of the negative log-likelihood. Therefore, we have

$$W_{\text{tgt}} - W_0 \approx - J(W_0)^{-1} G. \tag{16}$$

After obtaining an estimator of $W_{\text{tgt}} - W_0$ and computing $A_0$ via Theorem 4.1, combining with (44), the initialization for $B$ is

$$B_0 = A_0^\top (W_{\text{tgt}} - W_0), \tag{17}$$

so that $W_0 + A_0 B_0$ coincides with the projection $W_{\text{tgt}}^{\text{proj}}$.

## 4.2. Standard LoRA Case

We next consider the standard LoRA case where $W \in \mathbb{R}^{d_1 \times d_2}$. In this setting, the estimator is given by $\hat{W} = W_0 + A B$, where $A \in \mathbb{R}^{d_1 \times r}$ denotes a fixed column-orthogonal matrix, and $B \in \mathbb{R}^{r \times d_2}$ is a trainable matrix.

**Theorem 4.2.** *(proved in Appendix C.2) In LoRA fine-tuning from $P_{X;W_0}$ to $P_{X;W_{\text{tgt}}}$, we assume $W_0, W_{\text{tgt}} \in \mathbb{R}^{d_1 \times d_2}$ and $\|W_{\text{tgt}} - W_0\|_F = O\left(\frac{1}{\sqrt{N}}\right)$. Then, the optimal initialization of the LoRA matrix $A$ is given by*

$$A_0^* = \arg\min_A \text{tr}\left(A^\top \Omega A\right), \tag{18}$$

*and from Section 2.3 we know that the $r$ column vectors of $A$ correspond to the eigenvectors of $\Omega$ associated with its $r$ smallest eigenvalues. $\Omega$ is the **Initialization Guidance Matrix** given by*

$$\Omega^{d_1 \times d_1} =$$
$$\sum_{i=1}^{d_2} \frac{J(W_0)_{[i]}^{-1}}{N} - (W_{\text{tgt}} - W_0)_{(:,i)} (W_{\text{tgt}} - W_0)_{(:,i)}^\top, \tag{19}$$

*where the subscript $(:, i)$ denotes the $i$-th column of a matrix. It is important to note that the columns of $W$ are not independent, and therefore the Fisher information matrix should be defined with respect to the entire parameter matrix rather than computed separately for each column. Specifically, $J(W_0)_{[i]}^{-1}$ denotes the $i$-th $d_1 \times d_1$ diagonal block of the inverse Fisher matrix $J(\text{vec}(W_0))^{-1}$, i.e.,*

$$J(W_0)_{[i]}^{-1} \triangleq J(\text{vec}(W_0))_{((i-1)d_1+1:id_1,\ (i-1)d_1+1:id_1)}^{-1}, \tag{20}$$

*where $\text{vec}(\cdot)$ denotes the column-wise vectorization operator that flattens $W$ into a vector.*

Similar to (16), the approximation of $W_{\text{tgt}} - W_0$ is

$$(W_{\text{tgt}} - W_0)_{(:,i)} \approx - J(W_0)_{[i]}^{-1} G_{(:,i)}, \tag{21}$$

where $G^{d_1 \times d_2} = \nabla \mathcal{L}_{\text{tgt}}(W_0)$. Moreover, the initialization method is the same as (17).

*Remark* 4.3. To relate our method to prior gradient-based fine-tuning and to justify the optimality of our initialization, we present the following observation. Both LoRA-GA and LoRA-One essentially perform a direct singular value decomposition (SVD) on the gradient. If we simplify our approach by (i) discarding the first term in the Initialization Guidance Matrix and (ii) directly using the negative gradient to replace the right side in Eq. (21), our theoretical result in Theorem 4.2 reduces to

$$A_0^* = \arg\min_A - \text{tr}\left(A^\top G G^\top A\right). \tag{22}$$

In other words, the initialization of $A$ corresponds to the leading $r$ eigenvectors of $GG^\top$, which are equivalent to the top $r$ left singular vectors of $G$. Thus, our method in this degenerate form coincides with the strategies adopted in LoRA-GA and LoRA-One. From this analysis, two advantages of our theoretical results become evident: (1) the first term of the Initialization Guidance Matrix explicitly models the estimation variance induced by the stochasticity of training samples, a factor overlooked in prior studies; and (2) the estimation in (21), compared with using raw gradients alone, incorporates the Fisher information matrix to account for the anisotropy of the model.

## 4.3. Algorithm

Grounded in Sections 4.1 and 4.2, we introduce LoRA-DA, which translates our theory into a practical LoRA initialization scheme. Specifically, among the $N$ available target samples, LoRA-DA requires only a small subset of target samples $\mathcal{S}$ to estimate the necessary statistics for initialization, making it lightweight. This design is motivated by prior work suggesting that, in practice, relatively few samples can already provide useful estimates of the Fisher matrix and gradient statistics (Guo et al., 2024; Zhang et al., 2025b). Using this subset, we compute both the gradient and the Fisher matrix. The Fisher matrix is computed using the **K-FAC** (Martens & Grosse, 2015), a scalable method that approximates the Fisher as a Kronecker product of smaller matrices formed by the layer input vectors and the backpropagated gradients. Moreover, for computing the eigenvalues and eigenvectors required in the initialization of $A_0$, we employ the **LOBPCG** algorithm (Knyazev, 2001). The detailed procedure of LoRA-DA is shown in Algorithm 1. Although LoRA-DA is designed for the standard LoRA framework, we discuss in the Appendix E how this initialization can be extended to other LoRA-style PEFT methods.

Appendix D shows that our algorithm introduces no significant memory overhead compared with gradient-based methods. Moreover, we note that the time cost of LOBPCG scales well to large LLMs. As shown in Algorithm 1 and Theorem 4.2, the decomposition is performed independently for each LoRA layer. For a layer of size $d_1 \times d_2$, LOBPCG

---

**Algorithm 1** LoRA-DA for one specific layer

---

**Input:** Pre-trained weight $\boldsymbol{W}_0^{d_1 \times d_2}$, total target data size $N$, a small subset of target data $\mathcal{S} = \{(z^j, y^j)\}_{j=1}^{|\mathcal{S}|}$, LoRA rank $r$,
loss function $\ell$           // data size $|\mathcal{S}| \ll N$

**Initialize:**

1:   $\boldsymbol{G}^{d_1 \times d_2} \leftarrow \frac{1}{|\mathcal{S}|} \sum\limits_{(z^j, y^j) \in \mathcal{S}} \nabla_{\boldsymbol{W}} \ell(\boldsymbol{W}_0; z^j, y^j)$

2:   $\boldsymbol{Z}_{\text{fisher}}^{d_1 \times d_1} \leftarrow \frac{1}{|\mathcal{S}|} \sum\limits_{(z^j, y^j) \in \mathcal{S}} z^j {z^j}^\top$

3:   $\boldsymbol{Y}_{\text{fisher}}^{d_2 \times d_2} \leftarrow \frac{1}{|\mathcal{S}|} \sum\limits_{(z^j, y^j) \in \mathcal{S}} \nabla_y \ell(\boldsymbol{W}_0; z^j, y^j) \nabla_y \ell(\boldsymbol{W}_0; z^j, y^j)^\top$

4:   **for** $i = 1, \dots, d_2$ **do**

5:      $\boldsymbol{J}(\boldsymbol{W}_0)_{[i]}^{-1} \leftarrow \boldsymbol{Z}_{\text{fisher}}^{-1} \times [\boldsymbol{Y}_{\text{fisher}}^{-1}]_{(i,i)}$                 // $[\boldsymbol{Y}_{\text{fisher}}^{-1}]_{(i,i)}$ is the $(i,i)$ entry of $\boldsymbol{Y}_{\text{fisher}}^{-1}$.

6:      $(\boldsymbol{W}_{\text{tgt}} - \boldsymbol{W}_0)_{(:,i)} \leftarrow -\boldsymbol{J}(\boldsymbol{W}_0)_{[i]}^{-1} G_{(:,i)}$

7:   **end for**

8:   $\boldsymbol{A}_0 \leftarrow \arg\min_{\boldsymbol{A}} \operatorname{tr}\left( \boldsymbol{A}^\top \left( \sum_{i=1}^{d_2} \frac{\boldsymbol{J}(\boldsymbol{W}_0)_{[i]}^{-1}}{N} - \sum_{i=1}^{d_2} (\boldsymbol{W}_{\text{tgt}} - \boldsymbol{W}_0)_{(:,i)} (\boldsymbol{W}_{\text{tgt}} - \boldsymbol{W}_0)_{(:,i)}^\top \right) \boldsymbol{A} \right)$

9:   $\boldsymbol{B}_0 \leftarrow \boldsymbol{A}_0^\top (\boldsymbol{W}_{\text{tgt}} - \boldsymbol{W}_0)$

**Return:** $\boldsymbol{A}_0, \boldsymbol{B}_0$

---

operates only on the $d_1 \times d_1$ initialization guidance matrix $\boldsymbol{\Omega}$. Let $T_{\text{LOBPCG}}$ denote the number of iterations (typically a small constant, e.g., 10). The per-layer computational cost is $O(T_{\text{LOBPCG}} d_1^2 r)$. Importantly, as model size increases, the growth in parameters mainly comes from adding more layers or widening feedforward blocks, rather than increasing the per-layer input dimension $d_1$. Thus, $d_1$ does not scale with the total parameter count, and the per-layer cost remains essentially constant. Consequently, the overall eigenvalue computation cost grows only *linearly* with model depth.

## 5. Experiments

In this section, we conduct experiments to evaluate LoRA-DA against representative initialization strategies for LoRA across several NLP benchmarks. All experiments are performed on eight NVIDIA A800 GPUs, unless stated otherwise. We use 256 target samples by default to estimate the required statistics, and Appendix J shows that Fisher estimation remains stable at this sample size. Our baselines consist of vanilla LoRA (Hu et al., 2022) and its initialization variants with original structure: the data-agnostic methods PiSSA (Meng et al., 2024) and MiLoRA (Wang et al., 2025), and the gradient-based method LoRA-One (Zhang et al., 2025b). We do not include other PEFT methods that modify the original LoRA structure, despite their initialization modules, in order to maintain fairness in comparison. We provide the hyperparameter settings in the Appendix K.

### 5.1. Natural Language Understanding (NLU) Tasks

We adopt *commonsense reasoning* as a representative NLU task. Specifically, we fine-tune **LLaMA 2–7B** (Touvron et al., 2023) on all samples from **Commonsense170K** (Hu

et al., 2023), and evaluate on eight widely used benchmarks: **BoolQ** (Clark et al., 2019), **PIQA** (Bisk et al., 2020), **SIQA** (Sap et al., 2019), **HellaSwag** (Zellers et al., 2019), **WinoGrande** (Sakaguchi et al., 2021), **ARC-e** and **ARC-c** (Clark et al., 2018), and **OBQA** (Mihaylov et al., 2018). The task is formulated as a *multiple-choice problem*, and we report **accuracy (%)** on all test sets using the last checkpoint.

As shown in Table 1, LoRA-DA outperforms existing LoRA initialization methods in terms of average performance and across most benchmarks. On average, LoRA-DA attains 84.3%, exceeding the prior state-of-the-art MiLoRA (84.0%) with a margin of 0.3 percentage points. In particular, LoRA-DA achieves top performance on six out of eight benchmarks, demonstrating its robustness and consistency across diverse reasoning tasks.

### 5.2. Natural Language Generation (NLG) Tasks

We use *mathematical reasoning* as a representative NLG task, which requires models to generate a complete reasoning process and a final answer. Concretely, we fine-tune **LLaMA 2-7B** and **LLaMA 2-13B** (Touvron et al., 2023) on 100K samples from **MetaMathQA** (Yu et al., 2023), and evaluate on the official test sets of **GSM8K** (Cobbe et al., 2021) and **MATH** (Hendrycks et al., 2021). Unless otherwise specified, results are reported from the last checkpoint, and performance is measured by the **Exact Match (EM)** ratio between predictions and ground-truth answers.

As shown in Table 2 and Table 3, LoRA-DA achieves clear improvements over existing initialization strategies on mathematical reasoning tasks on both LLaMA 2-7B and LLaMA 2-13B settings. For the LLaMA 2-7B setting in

*Table 1.* Performance on Commonsense Reasoning Benchmarks.

| Model | PEFT | BoolQ | PIQA | SIQA | HellaSwag | WinoGrande | ARC-e | ARC-c | OBQA | Avg. |
|---|---|---|---|---|---|---|---|---|---|---|
| ChatGPT | – | 73.1 | 85.4 | 68.5 | 78.5 | 66.1 | 89.8 | 79.9 | 74.8 | 77.0 |
| | LoRA | 73.2 | 85.8 | 81.8 | 94.9 | 86.0 | 74.7 | 88.7 | 86.0 | 83.9 |
| | PiSSA | 72.5 | 85.2 | 81.9 | 94.2 | 86.7 | 73.7 | 87.1 | **87.0** | 83.5 |
| LLaMA 2-7B | MiLoRA | 73.1 | 85.3 | 81.8 | 95.1 | 86.3 | 75.3 | 88.6 | 86.8 | 84.0 |
| | LoRA-One | 72.8 | 85.3 | 81.9 | **95.2** | 85.6 | 74.9 | **88.8** | 86.4 | 83.9 |
| | LoRA-DA | **73.2** | **86.0** | **82.4** | **95.2** | **87.1** | **75.7** | 88.7 | 86.4 | **84.3** |

*Table 2.* Performance on Math Reasoning Benchmarks (LLaMA 2-7B).

| Method | GSM8K | MATH | Avg. |
|---|---|---|---|
| LoRA | 53.1 | 8.3 | 30.7 |
| PiSSA | 53.2 | 8.2 | 30.7 |
| MiLoRA | 52.9 | 8.3 | 30.6 |
| LoRA-One | 53.7 | 8.5 | 31.1 |
| LoRA-DA | **55.0** | **9.2** | **32.1** |

*Table 3.* Performance on Math Reasoning Benchmarks (LLaMA 2-13B).

| Method | GSM8K | MATH | Avg. |
|---|---|---|---|
| LoRA | 64.3 | 11.1 | 37.7 |
| PiSSA | 65.8 | 11.9 | 38.9 |
| MiLoRA | 64.6 | 12.2 | 38.4 |
| LoRA-One | 65.5 | 11.3 | 38.4 |
| LoRA-DA | **66.4** | **12.8** | **39.6** |

*Table 4.* Performance on Math Reasoning Benchmarks with frozen-$A$ setting (LLaMA 2-7B).

| Method | GSM8K |
|---|---|
| LoRA-FA | 41.5 |
| LoRA-DA | **49.4** |

Table 2, LoRA-DA attains the highest accuracy on both GSM8K and MATH, with accuracies of 55.0% and 9.2%, respectively. Compared to the strongest baseline LoRA-One, LoRA-DA improves the average accuracy by 1.0 percentage points. These results show that LoRA-DA initialization is effective not only for NLU tasks but also for NLG tasks. For the LLaMA-2-13B setting in Table 3, LoRA-DA also attains the highest accuracy on both GSM8K and MATH, with accuracies of 66.4% and 12.8%, respectively, demonstrating its effectiveness when applied to larger model scales. We also conduct a comparative experiment under the frozen-$A$ setting of LoRA-FA, as shown in Table 4. The results demonstrate that our initialization theory remains effective even when $A$ is frozen.

### 5.3. Performance Over Training Steps

We further analyze the optimization dynamics by tracking loss, gradient norm, and evaluation accuracy throughout training on GSM8K in Fig. 2. Compared with vanilla LoRA, LoRA-One achieves faster reduction of loss and higher accuracy in the earliest steps, since its gradient-only initialization closely aligns with the steepest descent direction. Interestingly, LoRA-DA starts slightly behind LoRA-One at the beginning. This behavior arises because it better accounts for sample stochasticity and parameter-space anisotropy. Its early steps appear more conservative—not because the method fails to find good directions, but because it prioritizes stability over immediate descent, leading to more reliable convergence in later stages. As training proceeds, LoRA-DA exhibits more stable gradient norms, faster overall convergence, and higher final accuracy. These results confirm that although gradient-only methods may appear

favorable in the short term, variance-aware initialization ultimately provides more robust optimization and superior long-run performance.

We provide another visualization of the descent trajectory of LoRA-DA in Fig. 3. Unlike full fine-tuning and vanilla LoRA that follow the raw gradient direction, LoRA-DA initialization uses a Fisher–gradient in the bias term to preserve anisotropy and Fisher information in the variance term to capture sampling-induced stochasticity. As a result, the descent trajectory of LoRA-DA does not coincide with that of full fine-tuning at the beginning. The updates of LoRA-DA appear more conservative because it prioritizes stability over immediate descent, thereby guiding the optimizer toward a more efficient convergence path than vanilla LoRA.

### 5.4. Experiments on Various Ranks

We further evaluate LoRA-DA under different ranks $r \in \{1, 2, 4, 8, 16\}$ on GSM8K, and compare it with LoRA and LoRA-One in Figure 4. Across all settings, LoRA-DA consistently outperforms both baselines, confirming its robustness. Notably, the advantage of LoRA-DA is more pronounced at smaller ranks. This can be explained by the fact that when the rank is severely limited, the choice of descent directions becomes particularly critical: a suboptimal initialization may constrain the model to an ineffective subspace that cannot be recovered during training. By contrast, LoRA-DA leverages Fisher-gradient and variance term to identify more informative low-rank directions, which substantially improves optimization in the low-rank regime. As the rank increases, the subspace coverage grows and the relative gap between methods narrows, yet LoRA-DA still

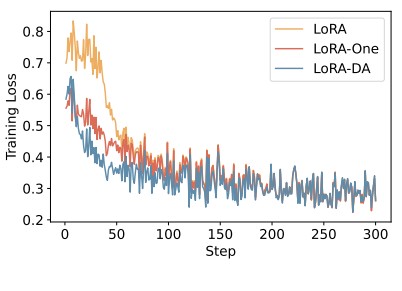

*(a)* Loss over steps.

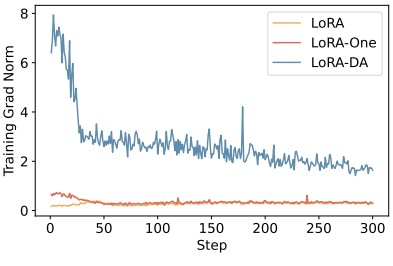

*(b)* Gradient norm over steps.

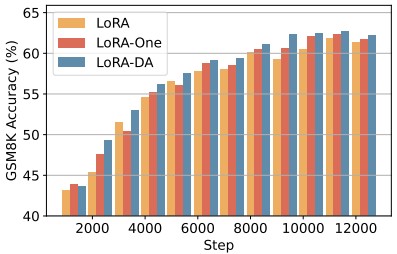

*(c)* Accuracy over steps.

*Figure 2.* The loss, grad norm, and evaluation accuracy on GSM8K over the training steps of LoRA (indicated in yellow), LoRA-One (in red), and LoRA-DA (in blue)

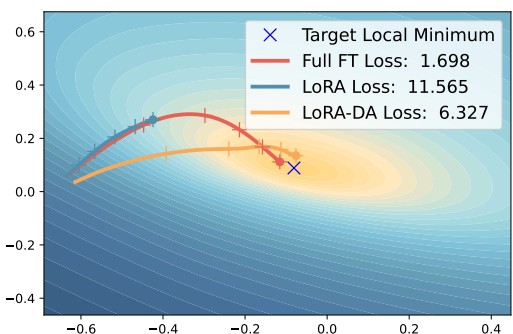

*Figure 3.* **Descent trajectory of LoRA-DA.** We follow the visualization setup in (Meng et al., 2024). Pre-training is conducted on 10,000 odd-numbered images from the MNIST dataset, followed by fine-tuning on 1,000 even-numbered images. The LoRA rank is set to 4, and the learning rate is set to $5 \times 10^{-4}$.

achieves the best performance across all ranks.

### 5.5. Initialization Overhead

Table 6 summarizes the initialization time, training time, and total time of LoRA-DA under different ranks on the GSM8K task. Here, the training time corresponds to the fine-tuning time shared by all LoRA-based methods, including the baselines, while the initialization time reflects the additional overhead introduced by LoRA-DA. Notably, initialization accounts for only about 6% of the total time, indicating that the extra cost of LoRA-DA remains small. Moreover, the initialization time remains stable across different ranks, suggesting that LoRA-DA scales well under varying rank settings.

### 5.6. Ablation Study

Table 7 reports the ablation study on GSM8K. Our method consists of two components: bias and variance. Removing the bias term (LoRA-DA w/o bias) or the variance term (LoRA-DA w/o var) both lead to slight drops in performance. Furthermore, LoRA-DA w/o var&fisher, which estimates the bias term using plain gradients instead of

Fisher-gradient, performs slightly worse than LoRA-DA w/o var. The full method (LoRA-DA) achieves the best results, showing that both components are essential for optimal performance. We further provide a layer-wise ablation in Appendix I, showing that LoRA-DA brings larger gains when applied to FFN layers, while also yielding consistent improvements on attention layers.

## 6. Related Work

PEFT methods fall into three categories: *adapter-based* (Houlsby et al., 2019; He et al., 2021); *prompt-based* (Lester et al., 2021; Razdaibiedina et al., 2023); and *LoRA-based* (Hu et al., 2022) methods. LoRA variants such as AdaLoRA (Zhang et al., 2023b), DoRA (Liu et al., 2024), and VeRA (Kopiczko et al., 2024) improve efficiency or stability, but most rely on random initialization, leading to slow warm-up and possible suboptimal convergence. Principled initialization has thus been explored. PiSSA (Meng et al., 2024) and MiLoRA (Wang et al., 2025) are data-agnostic, based on singular components of pre-trained weights, whereas data-aware methods leverage target samples, typically by using early gradients as in LoRA-GA (Wang et al., 2024a) and LoRA-One (Zhang et al., 2025b). However, their single-step reliance limits effectiveness. Beyond initialization, LoRA-Pro (Wang et al., 2024b) re-weights low-rank gradients during fine-tuning to better approximate full-model updates. We provide an extended version of the related work in the Appendix B.

## 7. Conclusion

We theoretically derive a data-aware initialization method for LoRA via asymptotic analysis. The proposed formulation decomposes the expected estimation error into a variance term characterized by the Fisher information and an anisotropy-aware bias term captured by the Fisher-gradient, and combines them to construct the initialization subspace. Based on this theory, we propose LoRA-DA, which uses a small set of target samples to compute LoRA initialization. On NLU and NLG benchmarks, LoRA-DA improves final

accuracy over prior LoRA initializations. Supplementary studies show faster and more stable convergence, robustness across ranks, and only a small initialization overhead.

## Acknowledgements

This work is supported in part by National Key R&D Program of China under Grant 2021YFA0715202 and the National Natural Science Foundation of China under Grants 62571296.

## Impact Statement

This paper presents work whose goal is to advance the field of Machine Learning. There are many potential societal consequences of our work, none which we feel must be specifically highlighted here.

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

# Appendix

## A. Notations

*Table 5.* Notation used in this paper.

| Symbol | Meaning | Shape / Type |
|---|---|---|
| $P_{X,\boldsymbol{W}_0}$ | Pre-trained model (data distribution) | |
| $P_{X,\boldsymbol{W}_{\text{tgt}}}$ | Target model (fine-tuning distribution) | |
| $X$ | Joint variable/distribution of inputs and outputs | |
| $Z$ | Model inputs/features | |
| $Y$ | Model outputs/labels | |
| $\boldsymbol{W}_0$ | Pre-trained parameter matrix | $\mathbb{R}^{d_1 \times d_2}$ |
| $\boldsymbol{W}_{\text{tgt}}$ | Target task true parameter matrix | $\mathbb{R}^{d_1 \times d_2}$ |
| $\hat{\boldsymbol{W}}$ | Estimated parameter after fine-tuning | $\mathbb{R}^{d_1 \times d_2}$ |
| $\Delta \boldsymbol{W}$ | Parameter update (LoRA low-rank) | $\mathbb{R}^{d_1 \times d_2}$ |
| $\boldsymbol{A}$ | LoRA left matrix | $\mathbb{R}^{d_1 \times r}$ |
| $\boldsymbol{B}$ | LoRA right matrix | $\mathbb{R}^{r \times d_2}$ |
| $r$ | LoRA rank | $r \ll \min(d_1, d_2)$ |
| $d_1, d_2$ | Row/column dimensions of parameter matrix | |
| $\mathcal{L}_{\text{tgt}}(\boldsymbol{W})$ | Empirical loss on target domain | |
| $\nabla \mathcal{L}_{\text{tgt}}(\boldsymbol{W}_0)$ | Gradient at $\boldsymbol{W}_0$ | $\mathbb{R}^{d_1 \times d_2}$ |
| $\boldsymbol{G}$ | Gradient at $\boldsymbol{W}_0$ | $\mathbb{R}^{d_1 \times d_2}$ |
| $\boldsymbol{H}_0$ | Hessian at $\boldsymbol{W}_0$ | $\mathbb{R}^{(d_1 d_2) \times (d_1 d_2)}$ |
| $\boldsymbol{J}(\cdot)$ | Fisher information matrix | |
| $\boldsymbol{J}(\cdot)^{-1}$ | Inverse Fisher information matrix | |
| $\boldsymbol{J}(\boldsymbol{W}_0)_{[i]}^{-1}$ | the $i$-th $d_1 \times d_1$ diagonal block of $\boldsymbol{J}(\text{vec}(\boldsymbol{W}_0))^{-1}$ | $\mathbb{R}^{d_1 \times d_1}$ |
| $\text{vec}(\cdot)$ | the column-wise vectorization operator that flattens matrix into a vector | |
| $N$ | Number of target-domain samples | $\mathbb{N}$ |
| $\boldsymbol{W}_{\text{tgt}}^{\text{proj}}$ | Projection of $\boldsymbol{W}_{\text{tgt}}$ onto $W_0 + \{AB\}$ | $\mathbb{R}^{d_1 \times d_2}$ |
| $\boldsymbol{P} = \boldsymbol{A}\boldsymbol{A}^\top$ | Orthogonal projector onto subspace spanned by $A$ | $\mathbb{R}^{d_1 \times d_1}$ |
| $\boldsymbol{Z}_{\text{fisher}}$ | K-FAC left factor (input second moments) | $\mathbb{R}^{d_1 \times d_1}$ |
| $\boldsymbol{Y}_{\text{fisher}}$ | K-FAC right factor (output-gradient second moments) | $\mathbb{R}^{d_2 \times d_2}$ |
| $\boldsymbol{A}_0$ | Initialized $A$ | $\mathbb{R}^{d_1 \times r}$ |
| $\boldsymbol{B}_0$ | Initialized $B$ | $\mathbb{R}^{r \times d_2}$ |
| $\mathcal{S}$ | Small target-domain sample set | |
| $\boldsymbol{\Omega}$ | Initialization Guidance Matrix | $\mathbb{R}^{d_1 \times d_1}$ |
| $U, \boldsymbol{\Lambda}$ | Eigenvectors/eigenvalues (EVD) | |
| $\boldsymbol{u}_i$ | $i$-th eigenvector | |
| $\lambda_i$ | $i$-th eigenvalue (ascending) | |
| $(:, i)$ | The $i$-th column of a matrix (colon indexing) | |
| $(i, i)$ | The $(i, i)$-th (diagonal) entry | |
| $\|\cdot\|_F$ | Frobenius norm | |
| $\text{tr}(\cdot)$ | Trace operator | |

# B. Extended Related Work

## B.1. Parameter-Efficient Fine-Tuning (PEFT)

PEFT aims to reduce the computational and memory overhead of adapting large pre-trained models, while still achieving performance comparable to full model fine-tuning. Existing PEFT techniques can be broadly divided into three categories, namely *adapter-based*, *prompt-based*, and *LoRA-based* methods. **Adapter-based methods** (Houlsby et al., 2019; He et al., 2021; Mahabadi et al., 2021)insert small trainable modules into frozen layers of the backbone network, enabling efficient task adaptation without updating the full set of parameters. **Prompt-based methods** (Lester et al., 2021; Razdaibiedina et al., 2023; Wang et al., 2023b) optimize additional continuous tokens (soft prompts) or prefix vectors that steer the behavior of the pre-trained model. These approaches enable highly lightweight adaptation with minimal trainable parameters. However, their performance is often sensitive to initialization and may vary considerably across different tasks. The above two categories, by altering either the architecture or the input representation, tend to introduce extra inference overhead compared to the backbone model.

## B.2. Low-Rank Adaptation (LoRA) and its Variants and Initialization

The third category, **LoRA-based methods**, performs parameter-efficient fine-tuning by constraining weight updates to a low-rank decomposition. Instead of directly updating the full parameter matrix, LoRA reparameterizes the update as the product of two low-rank matrices, which drastically reduces the number of trainable parameters while enabling the merged weights to be seamlessly integrated into the backbone at inference time without additional latency (Hu et al., 2022). For instance, LoRA-FA (Zhang et al., 2023a) introduces parameter freezing to reduce memory usage during adaptation; AdaLoRA (Zhang et al., 2023b) dynamically allocates rank across layers according to their importance, improving overall parameter efficiency; DoRA (Liu et al., 2024) decomposes pre-trained weights into magnitude and direction, applying low-rank updates only to the directional component for greater stability and expressiveness; VeRA (Kopiczko et al., 2024) re-parameterizes LoRA by introducing shared low-rank vectors across layers, thereby reducing the number of trainable parameters while maintaining competitive performance. However, LoRA and many of its variants typically adopt random initialization for the low-rank matrix $A$ and zero initialization for $B$. Such uninformed initialization introduces two main drawbacks. First, the training process tends to progress slowly in the early stages, as the update directions are not aligned with informative subspaces. Second, the optimization may converge to suboptimal solutions, since the imposed low-rank structure restricts the search space to poorly initialized directions.

To overcome these limitations, several works propose more principled initialization strategies. Some work investigates LoRA's two zero-start initialization schemes: initializing $A$ with $B = 0$ versus initializing $B$ with $A = 0$, and demonstrates that $A$-initialization is more effective (Hayou et al., 2024). PiSSA (Meng et al., 2024) initializes the low-rank matrices $A$ and $B$ using the principal singular components of the pre-trained weights, while the remaining components are used to initialize the residual weights $W$. In contrast, MiLoRA (Wang et al., 2025) leverages minor singular components for initializing $A$ and $B$, thereby preserving dominant directions of the pre-trained model and exploiting underutilized subspaces for adaptation. Early approaches of this kind are generally *data-agnostic*, as they rely solely on the weight structure. More recently, *data-aware* methods have been explored, which explicitly incorporate information from the target task. For example, LoRA-GA (Wang et al., 2024a) employs gradient alignment to initialize the low-rank subspace, and LoRA-One (Zhang et al., 2025b) derives its initialization by decomposing the fine-tuning gradient at the first optimization step. However, most existing data-aware approaches rely on such a direct decomposition of the first-step gradient. Since model performance after a single update step is typically unsatisfactory, initialization strategies grounded solely in this early gradient can be limited in effectiveness. In addition, several variants leverage gradients to assist training but are not directly related to initialization. For instance, LoRA-Pro (Wang et al., 2024b) improves LoRA by strategically re-weighting the gradients of the low-rank matrices during fine-tuning, such that their product produces a low-rank update that more faithfully approximates the full-model gradient, thereby narrowing the performance gap to full fine-tuning.

## C. Proofs

### C.1. Proof of Theorem 4.1

We assume that the orthogonal projection of $\boldsymbol{W}_{\text{tgt}}$ onto the LoRA subspace is denoted by $\boldsymbol{W}_{\text{tgt}}^{\text{proj}}$, as shown in Figure 1. Under this assumption, the training objective can be decomposed as

$$\mathbb{E}\left[\left\|\hat{\boldsymbol{W}} - \boldsymbol{W}_{\text{tgt}}\right\|_F^2\right] = \mathbb{E}\left[\left\|\hat{\boldsymbol{W}} - \boldsymbol{W}_{\text{tgt}}^{\text{proj}}\right\|_F^2\right] + \left\|\boldsymbol{W}_{\text{tgt}}^{\text{proj}} - \boldsymbol{W}_{\text{tgt}}\right\|_F^2, \tag{23}$$

where the first term represents the expected estimation variance within the subspace, and the second term corresponds to the bias arising from the distance between $\boldsymbol{W}_{\text{tgt}}$ and the LoRA subspace.

For the first term of (23), by Section 2.2, we know that if $\hat{\boldsymbol{W}}$ is not restricted to the LoRA form, the training variance is given by $\frac{\text{tr}\left(\boldsymbol{J}(\boldsymbol{W}_{\text{tgt}})^{-1}\right)}{N}$. In the presence of the LoRA structure with a fixed orthonormal basis $\boldsymbol{A}$, this variance is no longer full, but have a upper bound which is the projection of the unconstrained variance onto the subspace spanned by $\boldsymbol{A}$.

**Lemma C.1.** *Let $\{x_i\}_{i=1}^N$ be i.i.d. samples drawn from the target distribution $P_{X;\boldsymbol{W}_{\text{tgt}}}$. Let $\hat{\boldsymbol{W}}$ denote the maximum likelihood estimator constrained to the subspace $\mathcal{V} \triangleq \left\{\boldsymbol{W}_0 + \boldsymbol{A}\boldsymbol{B} \middle| \boldsymbol{A} \in \mathbb{R}^{d_1 \times r}, \boldsymbol{B} \in \mathbb{R}^{r \times 1}\right\}$, i.e.,*

$$\hat{\boldsymbol{W}} = \arg\max_{\boldsymbol{W} \in \mathcal{V}} \frac{1}{N} \sum_{i=1}^N \log P_{X,\boldsymbol{W}}(x_i). \tag{24}$$

*where $\boldsymbol{A} \in \mathbb{R}^{d_1 \times r}$ is fixed with orthonormal columns and $\boldsymbol{B} \in \mathbb{R}^{r \times 1}$ is the variable. Let $\boldsymbol{W}_{\text{tgt}}^{\text{proj}}$ denote the projection of $\boldsymbol{W}_{\text{tgt}}$ onto this subspace. Then, we have*

$$\mathbb{E}\left[\left\|\hat{\boldsymbol{W}} - \boldsymbol{W}_{\text{tgt}}^{\text{proj}}\right\|_F^2\right] \le \frac{\text{tr}\left(\boldsymbol{A}^\top \boldsymbol{J}(\boldsymbol{W}_{\text{tgt}})^{-1} \boldsymbol{A}\right)}{N} + o\left(\frac{1}{N}\right). \tag{25}$$

*Proof.* According to prior analyses of constrained maximum likelihood estimation (Aitchison & Silvey, 1958; Geyer, 1994), we have the following auxiliary lemma. However, the notation in the two cited references differs substantially from ours, making a direct adaptation unclear. While we acknowledge that the following auxiliary lemma originates from these references and is not our novel contribution, to avoid ambiguity, we now provide a complete and self-contained proof.

**Auxiliary Lemma C.2.**

$$\mathbb{E}\left[\left\|\hat{\boldsymbol{W}} - \boldsymbol{W}_{\text{tgt}}^{\text{proj}}\right\|_F^2\right] = \frac{\text{tr}\left((\boldsymbol{A}^\top \boldsymbol{J}(\boldsymbol{W}_{\text{tgt}})\boldsymbol{A})^{-1}\right)}{N} + o\left(\frac{1}{N}\right). \tag{26}$$

*Proof.* There exists $\boldsymbol{B}^*$ such that

$$\boldsymbol{W}_{\text{tgt}}^{\text{proj}} = \boldsymbol{W}_0 + \boldsymbol{A}\,\boldsymbol{B}^*,$$

maximizes the likelihood for parameter $\boldsymbol{W}$ over the subspace $\mathcal{V}$, and the constrained MLE is denoted by

$$\hat{\boldsymbol{W}} = \boldsymbol{W}_0 + \boldsymbol{A}\hat{\boldsymbol{B}}.$$

Let $\boldsymbol{g}_B$ denotes the derivative of $\log P_{X;\boldsymbol{W}} = \log P_{X;\boldsymbol{W}_0 + \boldsymbol{A}\boldsymbol{B}}$ with respect to $\boldsymbol{B}$, and let $\boldsymbol{g}_W$ denotes the derivative of $\log P_{X;\boldsymbol{W}_0 + \boldsymbol{A}\boldsymbol{B}}$ with respect to $\boldsymbol{W}$. Using the chain rule on $\boldsymbol{W} = \boldsymbol{W}_0 + \boldsymbol{A}\boldsymbol{B}$ gives

$$\boldsymbol{g}_B = \boldsymbol{A}^\top \boldsymbol{g}_W.$$

Thus, we have

$$\boldsymbol{J}_B(\boldsymbol{B}^*) = \mathbb{E}[\boldsymbol{g}_B \boldsymbol{g}_B^\top] = \boldsymbol{A}^\top \mathbb{E}[\boldsymbol{g}_W \boldsymbol{g}_W^\top]\boldsymbol{A} = \boldsymbol{A}^\top \boldsymbol{J}(\boldsymbol{W}_{\text{tgt}}^{\text{proj}})\boldsymbol{A}.$$

From the MLE asymptotics in Section 2.2, we have

$$\sqrt{N}(\hat{\boldsymbol{B}} - \boldsymbol{B}^*) \to N(0, (\boldsymbol{A}^\top \boldsymbol{J}(\boldsymbol{W}_{\text{tgt}}^{\text{proj}})\boldsymbol{A})^{-1}).$$

Since

$$\hat{\boldsymbol{W}} - \boldsymbol{W}_{\text{tgt}}^{\text{proj}} = \boldsymbol{A}(\hat{\boldsymbol{B}} - \boldsymbol{B}^*),$$

we have

$$Cov(\hat{\boldsymbol{W}} - \boldsymbol{W}_{\text{tgt}}^{\text{proj}}) = \frac{1}{N}\boldsymbol{A}(\boldsymbol{A}^\top \boldsymbol{J}(\boldsymbol{W}_{\text{tgt}}^{\text{proj}})\boldsymbol{A})^{-1}\boldsymbol{A}^\top.$$

Combining the assumption $\|\boldsymbol{W}_{\text{tgt}} - \boldsymbol{W}_0\|_F = O\left(\frac{1}{\sqrt{N}}\right)$ with the positional relationship in Fig. 1 , we can gain $\|\boldsymbol{W}_{\text{tgt}} - \boldsymbol{W}_{\text{tgt}}^{\text{proj}}\|_F = O\left(\frac{1}{\sqrt{N}}\right)$. From the result in (Zhang et al., 2025a), we know that their Fisher matrix is also close, i.e., $\boldsymbol{J}(\boldsymbol{W}_{\text{tgt}}^{\text{proj}}) = \boldsymbol{J}(\boldsymbol{W}_{\text{tgt}}) + O(\frac{1}{\sqrt{N}})$, and a similar proof is also provided in (37). Thus, we have

$$Cov(\hat{\boldsymbol{W}} - \boldsymbol{W}_{\text{tgt}}^{\text{proj}}) = \frac{1}{N}\boldsymbol{A}\left(\boldsymbol{A}^\top\left(\boldsymbol{J}\left(\boldsymbol{W}_{\text{tgt}} + O\left(\frac{1}{\sqrt{N}}\right)\right)\right)\boldsymbol{A}\right)^{-1}\boldsymbol{A}^\top = \frac{1}{N}\boldsymbol{A}(\boldsymbol{A}^\top \boldsymbol{J}(\boldsymbol{W}_{\text{tgt}})\boldsymbol{A})^{-1}\boldsymbol{A}^\top + o(\tfrac{1}{N}).$$

Therefore, we have

$$\mathbb{E}[\|\hat{\boldsymbol{W}} - \boldsymbol{W}_{\text{tgt}}^{\text{proj}}\|_F^2] = \frac{1}{N}tr\left(\boldsymbol{A}(\boldsymbol{A}^\top \boldsymbol{J}(\boldsymbol{W}_{\text{tgt}})\boldsymbol{A})^{-1}\boldsymbol{A}^\top\right) + o(\tfrac{1}{N}).$$

Using $\boldsymbol{A}^\top \boldsymbol{A} = I_r$ and the cyclic trace identity, we can prove the (26) in Auxiliary Lemma C.2, i.e.,

$$\mathbb{E}[\|\hat{\boldsymbol{W}} - \boldsymbol{W}_{\text{tgt}}^{\text{proj}}\|_F^2] = \frac{tr\left((\boldsymbol{A}^\top \boldsymbol{J}(\boldsymbol{W}_{\text{tgt}})\boldsymbol{A})^{-1}\right)}{N} + o\left(\tfrac{1}{N}\right).$$

$\square$

However, the form of Auxiliary Lemma C.2 is less convenient for the subsequent analysis, and we therefore apply a relaxation. In the following, we will show that

$$\text{tr}\left((\boldsymbol{A}^\top \boldsymbol{J}(\boldsymbol{W}_{\text{tgt}})\boldsymbol{A})^{-1}\right) \le \text{tr}\left(\boldsymbol{A}^\top \boldsymbol{J}(\boldsymbol{W}_{\text{tgt}})^{-1}\boldsymbol{A}\right). \tag{27}$$

We recall a variational identity: for any symmetric positive definite matrix $\boldsymbol{M} \in \mathbb{R}^{d \times d}$ and vector $v \in \mathbb{R}^d$,

$$v^\top \boldsymbol{M}^{-1}v = \max_{x \in \mathbb{R}^d}\left(2v^\top x - x^\top \boldsymbol{M}x\right). \tag{28}$$

This follows from completing the square, with the maximum attained at $x^\star = \boldsymbol{M}^{-1}v$.

Now set $\boldsymbol{M} = \boldsymbol{J}(\boldsymbol{W}_{\text{tgt}})$ and $v = \boldsymbol{A}y$ for arbitrary $y \in \mathbb{R}^r$. By (28), we have

$$y^\top \boldsymbol{A}^\top \boldsymbol{J}(\boldsymbol{W}_{\text{tgt}})^{-1}\boldsymbol{A}y = \max_{x \in \mathbb{R}^d}\left(2y^\top \boldsymbol{A}^\top x - x^\top \boldsymbol{J}(\boldsymbol{W}_{\text{tgt}})x\right). \tag{29}$$

Restricting the maximization in (29) to the subspace spanned by the columns of $\boldsymbol{A}$ cannot increase the maximum. By writing $x = \boldsymbol{A}u$ with $u \in \mathbb{R}^r$, we obtain

$$y^\top \boldsymbol{A}^\top \boldsymbol{J}(\boldsymbol{W}_{\text{tgt}})^{-1}\boldsymbol{A}y \ge \max_{u \in \mathbb{R}^r}\left(2y^\top u - u^\top (\boldsymbol{A}^\top \boldsymbol{J}(\boldsymbol{W}_{\text{tgt}})\boldsymbol{A})u\right). \tag{30}$$

Applying (28) again with $\boldsymbol{M} = \boldsymbol{A}^\top \boldsymbol{J}(\boldsymbol{W}_{\text{tgt}})\boldsymbol{A}$ and $v = y$, it follows that

$$y^\top (\boldsymbol{A}^\top \boldsymbol{J}(\boldsymbol{W}_{\text{tgt}})\boldsymbol{A})^{-1}y = \max_{x \in \mathbb{R}^r}\left(2y^\top x - x^\top (\boldsymbol{A}^\top \boldsymbol{J}(\boldsymbol{W}_{\text{tgt}})\boldsymbol{A})x\right). \tag{31}$$

The right-hand side of (30) is equivalent to the right-hand side of (31). Thus, for all $y \in \mathbb{R}^r$,

$$y^\top \boldsymbol{A}^\top \boldsymbol{J}(\boldsymbol{W}_{\text{tgt}})^{-1}\boldsymbol{A}y \ge y^\top (\boldsymbol{A}^\top \boldsymbol{J}(\boldsymbol{W}_{\text{tgt}})\boldsymbol{A})^{-1}y, \tag{32}$$

which implies the Loewner order

$$(A^\top J(W_{\text{tgt}})A)^{-1} \preceq A^\top J(W_{\text{tgt}})^{-1}A. \tag{33}$$

Finally, since the trace is monotone with respect to the Loewner order on positive semidefinite matrices, we conclude

$$\text{tr}\big((A^\top J(W_{\text{tgt}})A)^{-1}\big) \leq \text{tr}\big(A^\top J(W_{\text{tgt}})^{-1}A\big). \tag{34}$$

Equality holds if and only if the unconstrained maximizer $x^\star = J(W_{\text{tgt}})^{-1}Ay$ always lies in the subspace spanned by the columns of $A$. This is equivalent to requiring that the subspace spanned by the columns of $A$ be invariant under $J(W_{\text{tgt}})$, for instance, when the columns of $A$ are eigenvectors of $J(W_{\text{tgt}})$, or when $J(W_{\text{tgt}})$ is a scalar multiple of the identity matrix.

Combining (34) and (26), we can prove (25).

$\square$

Given that $W_{\text{tgt}}, W_0$ are already assumed to be sufficiently close, we can use this, along with a Taylor expansion, to approximate the distance between their inverse Fisher information matrices. We conclude the following lemma.

**Lemma C.3.** *For $W_{\text{tgt}}$ and $W_0$ satisfy $\|W_{\text{tgt}} - W_0\|_F = O\left(\frac{1}{\sqrt{N}}\right)$, the discrepancy between $J(W_{\text{tgt}})^{-1}$ and $J(W_0)^{-1}$ is of the order $O\left(\frac{1}{\sqrt{N}}\right)$, i.e.,*

$$J(W_{\text{tgt}})^{-1} = J(W_0)^{-1} + O(\tfrac{1}{\sqrt{N}}). \tag{35}$$

*Proof.*

$$J(W_{\text{tgt}}) \tag{36}$$

$$= \mathbb{E}_{W_{\text{tgt}}}\left[\left(\frac{\partial}{\partial W}\log P_{X;W_{\text{tgt}}}\right)\left(\frac{\partial}{\partial W}\log P_{X;W_{\text{tgt}}}\right)^\top\right]$$

$$= \mathbb{E}_{W_{\text{tgt}}}\left[\left(\frac{\partial}{\partial W}\log P_{X;W_0} + \frac{\partial^2 \log P_{X;W_0}}{\partial W^2}(W_{\text{tgt}} - W_0) + O(\tfrac{1}{N})\right)\right.$$

$$\left.\left(\frac{\partial}{\partial W}\log P_{X;W_0} + \frac{\partial^2 \log P_{X;W_0}}{\partial W^2}(W_{\text{tgt}} - W_0) + O(\tfrac{1}{N})\right)^\top\right]$$

$$= \mathbb{E}_{W_{\text{tgt}}}\left[\left(\frac{\partial}{\partial W}\log P_{X;W_0}\right)\left(\frac{\partial}{\partial W}\log P_{X;W_0}\right)^\top\right] + O(\tfrac{1}{\sqrt{N}})$$

$$= \sum_{x\in\mathcal{X}} P_{X;W_{\text{tgt}}}(x)\left(\frac{\partial}{\partial W}\log P_{X;W_0}(x)\right)\left(\frac{\partial}{\partial W}\log P_{X;W_0}(x)\right)^\top + O(\tfrac{1}{\sqrt{N}})$$

$$= \sum_{x\in\mathcal{X}}\left(P_{X;W_0}(x) + \frac{\partial P_{X;W_0}}{\partial W}(W_{\text{tgt}} - W_0) + O(\tfrac{1}{N})\right)\left(\frac{\partial}{\partial W}\log P_{X;W_0}(x)\right)\left(\frac{\partial}{\partial W}\log P_{X;W_0}(x)\right)^\top + O(\tfrac{1}{\sqrt{N}})$$

$$= \sum_{x\in\mathcal{X}} P_{X;W_0}(x)\left(\frac{\partial}{\partial W}\log P_{X;W_0}(x)\right)\left(\frac{\partial}{\partial W}\log P_{X;W_0}(x)\right)^\top + O(\tfrac{1}{\sqrt{N}})$$

$$= J(W_0) + O(\tfrac{1}{\sqrt{N}}). \tag{37}$$

Assume that $J(W_0)$ is invertible with bounded condition number, and we have know that

$$J(W_{\text{tgt}}) = J(W_0) + O(\tfrac{1}{\sqrt{N}}). \tag{38}$$

By the resolvent identity (Horn & Johnson, 2012), we have

$$J(W_{\text{tgt}})^{-1} - J(W_0)^{-1} = J(W_{\text{tgt}})^{-1}\big(J(W_0) - J(W_{\text{tgt}})\big)J(W_0)^{-1}, \tag{39}$$

which can be easily proved by multiplying both sides of the equation on the left by $J(W_{\text{tgt}})$.

Taking norms on both sides yields

$$\left\| J(W_{\text{tgt}})^{-1} - J(W_0)^{-1} \right\| \leq \left\| J(W_{\text{tgt}})^{-1} \right\| \left\| J(W_{\text{tgt}}) - J(W_0) \right\| \left\| J(W_0)^{-1} \right\|. \tag{40}$$

We analyze the right-hand side of the inequality. Since the Fisher matrix and its inverse are both of constant order and $\|J(W_{\text{tgt}}) - J(W_0)\| = O(\frac{1}{\sqrt{N}})$, the right-hand side of the inequality is of order $O\left(\frac{1}{\sqrt{N}}\right)$. Therefore, from (40) we can prove the Lemma C.3, i.e.,

$$J(W_{\text{tgt}})^{-1} = J(W_0)^{-1} + O(\tfrac{1}{\sqrt{N}}). \tag{41}$$

$\square$

Combining Lemma C.3 and (25), we have

$$\mathbb{E}\left[\left\| \hat{W} - W_{\text{tgt}}^{\text{proj}} \right\|_F^2\right] \leq \frac{\text{tr}\left(A^\top \left(J(W_0)^{-1} + O(\frac{1}{\sqrt{N}})\right) A\right)}{N} = \frac{\text{tr}\left(A^\top J(W_0)^{-1} A\right)}{N} + o\left(\tfrac{1}{N}\right) \tag{42}$$

For the second term of (23), by the Pythagorean theorem in Euclidean space, we have

$$\left\| W_{\text{tgt}}^{\text{proj}} - W_{\text{tgt}} \right\|_F^2 = \|W_{\text{tgt}} - W_0\|_F^2 - \left\| W_{\text{tgt}}^{\text{proj}} - W_0 \right\|_F^2. \tag{43}$$

Since $A^\top A = I$, $AA^\top$ is the orthogonal projection matrix onto the column space of $A$. Therefore, as illustrated in Figure 1, the projection of the difference $W_{\text{tgt}} - W_0$ onto the subspace spanned by the columns of $A$ is

$$(W_{\text{tgt}} - W_0)^{\text{proj}} = AA^\top(W_{\text{tgt}} - W_0) = W_{\text{tgt}}^{\text{proj}} - W_0. \tag{44}$$

Moreover, we have

$$\begin{aligned}
\left\| W_{\text{tgt}}^{\text{proj}} - W_0 \right\|_F^2 &= \left\| AA^\top(W_{\text{tgt}} - W_0) \right\|_F^2 \\
&= \text{tr}\left((W_{\text{tgt}} - W_0)^\top AA^\top AA^\top(W_{\text{tgt}} - W_0)\right) \\
&= \text{tr}\left((W_{\text{tgt}} - W_0)^\top AA^\top(W_{\text{tgt}} - W_0)\right) \\
&= \left\| A^\top(W_{\text{tgt}} - W_0) \right\|_F^2.
\end{aligned} \tag{45}$$

Combining (43) and (45), we have

$$\left\| W_{\text{tgt}}^{\text{proj}} - W_{\text{tgt}} \right\|_F^2 = \|W_{\text{tgt}} - W_0\|_F^2 - \left\| A^\top(W_{\text{tgt}} - W_0) \right\|_F^2. \tag{46}$$

Combining (23) and (42) and (46), we have,

$$\begin{aligned}
\mathbb{E}\left[\left\| \hat{W} - W_{\text{tgt}} \right\|_F^2\right] \leq & \|W_{\text{tgt}} - W_0\|_F^2 \\
& + \text{tr}\left(A^\top \left(\frac{J(W_0)^{-1}}{N} - (W_{\text{tgt}} - W_0)(W_{\text{tgt}} - W_0)^\top\right) A\right) + o\left(\tfrac{1}{N}\right)
\end{aligned} \tag{47}$$

### C.2. Proof of Theorem 4.2

Similar to (23), the training objective can be decomposed as

$$\mathbb{E}\left[\left\| \hat{W} - W_{\text{tgt}} \right\|_F^2\right] = \mathbb{E}\left[\left\| \hat{W} - W_{\text{tgt}}^{\text{proj}} \right\|_F^2\right] + \left\| W_{\text{tgt}}^{\text{proj}} - W_{\text{tgt}} \right\|_F^2, \tag{48}$$

where the first term represents the expected estimation variance within the subspace, and the second term corresponds to the bias arising from the distance between $\boldsymbol{W}_{\text{tgt}}$ and the LoRA subspace.

For the first term of (48), similar to (42), we know that

$$\mathbb{E}\left[\left\|\hat{\boldsymbol{W}} - \boldsymbol{W}_{\text{tgt}}^{\text{proj}}\right\|_F^2\right] = \sum_{i=1}^{d_2} \mathbb{E}\left[\left\|(\hat{\boldsymbol{W}} - \boldsymbol{W}_{\text{tgt}}^{\text{proj}})_{(:,i)}\right\|_F^2\right]$$

$$\leq \sum_{i=1}^{d_2} \text{tr}\left(\frac{\boldsymbol{A}^\top \boldsymbol{J}(\boldsymbol{W}_0)_{[i]}^{-1} \boldsymbol{A}}{N}\right) + o\left(\tfrac{1}{N}\right), \tag{49}$$

where $\boldsymbol{J}(\boldsymbol{W}_0)_{[i]}^{-1}$ is denoted in Theorem 4.2.

For the second term of (48), similar to (46), we know that

$$\left\|\boldsymbol{W}_{\text{tgt}}^{\text{proj}} - \boldsymbol{W}_{\text{tgt}}\right\|_F^2 = \sum_{i=1}^{d_2} \left[\left\|(\boldsymbol{W}_{\text{tgt}}^{\text{proj}} - \boldsymbol{W}_{\text{tgt}})_{(:,i)}\right\|_F^2\right]$$

$$= \sum_{i=1}^{d_2} \left\|(\boldsymbol{W}_{\text{tgt}} - \boldsymbol{W}_0)_{(:,i)}\right\|_F^2 - \left\|\boldsymbol{A}^\top (\boldsymbol{W}_{\text{tgt}} - \boldsymbol{W}_0)_{(:,i)}\right\|_F^2 \tag{50}$$

Combining (48), (49) and (50), we have

$$\mathbb{E}\left[\left\|\hat{\boldsymbol{W}} - \boldsymbol{W}_{\text{tgt}}\right\|_F^2\right] \leq \sum_{i=1}^{d_2} \left\|(\boldsymbol{W}_{\text{tgt}} - \boldsymbol{W}_0)_{(:,i)}\right\|_F^2$$

$$+ \text{tr}\left(\boldsymbol{A}^\top \sum_{i=1}^{d_2} \left(\frac{\boldsymbol{J}(\boldsymbol{W}_0)_{[i]}^{-1}}{N} - (\boldsymbol{W}_{\text{tgt}} - \boldsymbol{W}_0)_{(:,i)} (\boldsymbol{W}_{\text{tgt}} - \boldsymbol{W}_0)_{(:,i)}^\top\right) \boldsymbol{A}\right) + o\left(\tfrac{1}{N}\right) \tag{51}$$

## D. Discussion on Space Complexity of the Algorithm

In our Algorithm 1, during the computation of statistics in Lines 1–3, we need to maintain $G \in \mathbb{R}^{d_1 \times d_2}$, $Y \in \mathbb{R}^{d_2 \times d_2}$, and $Z \in \mathbb{R}^{d_1 \times d_1}$. In fact, only the diagonal entries of $Y$ are required, which reduces its memory footprint to $\mathcal{O}(d_2)$. Moreover, $G$ and the pair of $Z, Y$ can be computed in two separate passes rather than stored simultaneously. Excluding the storage of the base weight $W_0$, which already requires $\mathcal{O}(d_1 d_2)$, the additional peak memory of our method is bounded by $\mathcal{O}(\max\{d_1^2, d_1 d_2\})$. In practical architectures such as LLaMA, LoRA is typically applied to attention and feed-forward projection layers where $d_1$ and $d_2$ are of comparable scale, so that $d_1^2$ and $d_1 d_2$ are of the same order. Hence, the peak memory usage of our algorithm is comparable to the gradient computation in LoRA-One which need a $\mathcal{O}(d_1 d_2)$ additional space to storage gradient. After these computations, all intermediate quantities can be offloaded from memory. At Line 7, we only need to maintain a $d_1 \times d_1$ coefficient matrix for the quadratic optimization, and the required statistics can be loaded in blocks from external storage. Furthermore, since we do not process all layers simultaneously, but instead compute the optimal LoRA initialization layer by layer in sequence, the overall peak memory usage remains bounded by the same order. **Therefore, our method achieves data-aware initialization without incurring higher space complexity than existing gradient-based approaches.**

## E. Extending LoRA-DA to Other LoRA-Style PEFT Methods

The Fisher-guided idea in LoRA-DA can extend naturally to other PEFT architectures, such as DoRA (Liu et al., 2024) and VeRA (Kopiczko et al., 2024). In LoRA-DA, we compute a Fisher-informed low-rank direction $A$ that captures the dominant curvature around $W_0$. This same $A$ can be directly reused in DoRA: DoRA separates the update into a magnitude term and a direction vector, and our Fisher-based $A$ provides exactly such an initialization for the direction component. Similarly, VeRA factorizes the update through a shared low-rank dictionary; the Fisher-guided $A$ can be used to initialize the dictionary atoms or the update subspace. Although LoRA-DA computes $A$ on a per-layer basis, this is not in conflict with VeRA's cross-layer sharing. VeRA constructs a shared low-rank dictionary across layers, and the Fisher-guided directions from LoRA-DA can be aggregated (e.g., via averaging, clustering, or PCA across layers) to form high-quality dictionary atoms. In this way, LoRA-DA supplies curvature-aware update directions that replace the random shared vectors used in VeRA, injecting task-aware structure into the VeRA parameterization.

# F. Accuracy Across Different Ranks

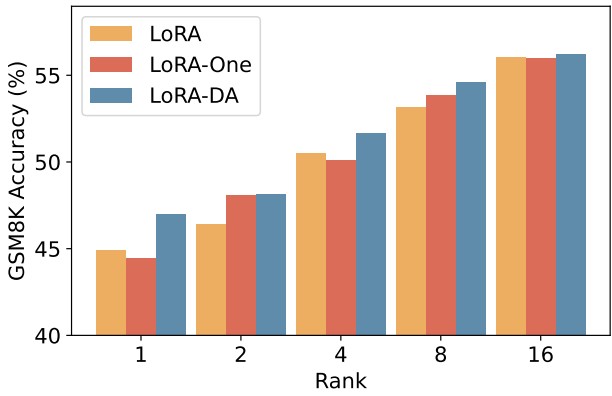

*Figure 4.* Accuracy of LoRA-DA across different ranks on the GSM8K task.

# G. Time Consumption Across Different Ranks

*Table 6.* Time consumption of LoRA-DA under different ranks on the GSM8K task, including initialization, training, and total time.

| Rank | 1 | 2 | 4 | 8 | 16 |
|---|---|---|---|---|---|
| Initialization Time | 2:03 | 2:04 | 2:04 | 2:05 | 2:06 |
| Training Time | 30:49 | 31:42 | 31:56 | 32:08 | 32:16 |
| Total Time | 32:52 | 33:46 | 34:00 | 34:13 | 34:22 |

# H. Ablation Experiments

*Table 7.* Ablation study of LoRA-DA on the GSM8K benchmark, evaluating the contributions of the bias and variance components.

| Method | GSM8K |
|---|---|
| LoRA-DA w/o bias | 53.6 |
| LoRA-DA w/o var | 53.3 |
| LoRA-DA w/o var&Fisher | 53.0 |
| LoRA-DA | **55.0** |

# I. Layer-wise Ablation

We further conduct a layer-wise ablation by applying LoRA-DA only to selected layers while keeping standard random initialization for the remaining layers. As shown in Table 8, the gains are more pronounced in FFN layers, while applying LoRA-DA to attention layers also yields smaller but consistent improvements.

*Table 8.* Layer-wise ablation of LoRA-DA on GSM8K. LoRA-DA is applied only to selected layer types, while the remaining layers use standard random initialization.

| Initialization Setting | GSM8K |
|---|---|
| LoRA (all random init) | 53.1 |
| LoRA-DA on attention only | 53.8 |
| LoRA-DA on FFN only | 54.3 |
| LoRA-DA on all layers | 55.0 |

## J. Fisher Estimation Stability

LQ-LoRA (Guo et al., 2024) shows that the Fisher matrix can be estimated from limited data without noticeably degrading downstream performance. To further validate this observation in our setting, we conduct an additional experiment on GSM8K with LLaMA-2-7B, where we vary the number of samples used for Fisher estimation during initialization. The results indicate that the Fisher matrix remains stable across a wide range of sample sizes, and that LoRA-DA achieves reliable update directions even with a small sample budget (256 samples in our default setting).

*Table 9.* Sensitivity to the number of samples used for Fisher estimation on GSM8K (LLaMA-2-7B).

| Fisher Samples | Accuracy (%) |
| --- | --- |
| LoRA-DA-4096 | 55.1 |
| LoRA-DA-1024 | 55.1 |
| LoRA-DA-256 | 55.0 |
| LoRA-DA-64 | 54.7 |
| LoRA-DA-16 | 54.5 |

## K. Hyperparameter Settings

To ensure a fair comparison among different low-rank adaptation methods, we used a unified hyperparameter configuration for LoRA-DA, LoRA, PiSSA, MiLoRA, and LoRA-One. This configuration was applied consistently to both NLG and NLU tasks, and the detailed hyperparameter settings are summarized in Table 10. We evaluated the models using the LLM-adapters framework (Hu et al., 2023). Notably, for LoRA-DA and LoRA-One, we pre-sampled 256 examples to estimate both the gradient and the Fisher information matrix during initialization.

*Table 10.* Hyperparameter settings for LoRA-DA and baseline methods.

| Hyperparameter | Value |
| --- | --- |
| LoRA Rank ($r$) | 8 |
| LoRA Alpha ($\alpha$) | 16 |
| LoRA Dropout | 0 |
| Target Modules (7B) | Q, K, V, O, Gate, Up, Down |
| Target Modules (13B) | Q, K, V |
| Sequence Length | 1024 |
| Batch Size | 32 |
| Gradient Samples | 256 |
| Learning Rate | $2 \times 10^{-4}$ |
| Weight Decay | 0 |
| Warmup Ratio | 0.03 |
| Warmup Steps | 100 |
| LR Scheduler | Cosine |
| Epochs (7B) | 1 |
| Epochs (13B) | 2 |
| Optimizer | AdamW |

## L. Reproducibility and Implementation Details

We have made efforts to ensure the reproducibility of our results. A detailed description of the proposed algorithm is provided in Algorithm 1, and all theoretical results are accompanied by complete proofs in the appendix. All datasets used are publicly available. We provide the hyperparameter settings in the Appendix K. The source code will be released upon publication to facilitate reproducibility and allow further verification of our results.

