# OpenReview forum: "LoRA-DA: Data-Aware Initialization for Low-Rank Adaptation via Asymptotic Analysis"
_ICML.cc/2026/Conference — ICML 2026 regular_

### Official Review · Reviewer_D9yC · 2026-03-07

**Soundness:** 3
**Presentation:** 3
**Significance:** 2
**Originality:** 2
**Overall Recommendation:** 4
**Confidence:** 3

**Summary:**

This paper proposes LoRA-DA, a data-aware initialization method for Low-Rank Adaptation (LoRA) in large language model fine-tuning. The method formulates LoRA initialization as minimizing the expected discrepancy between the fine-tuned parameters and the target task parameters under an asymptotic statistical framework. The authors derive an objective that decomposes the expected estimation error into bias and variance components, where the variance term is related to the Fisher information matrix and the bias term captures the discrepancy between pretrained parameters and target parameters.

Since the true target parameters are unknown, the method approximates the parameter displacement using a Fisher-gradient (natural-gradient–like) approximation and estimates curvature using K-FAC. This leads to a quadratic optimization problem whose solution corresponds to selecting eigenvectors associated with the smallest eigenvalues of a derived guidance matrix to determine the LoRA initialization subspace. Experiments are conducted on LLaMA-2 models (7B and 13B) across several natural language understanding and reasoning benchmarks, comparing the proposed approach with several existing LoRA initialization methods.

**Compliance With Llm Reviewing Policy:**

Affirmed.

**Final Justification:**

The paper proposes a theoretically motivated, data-aware initialization strategy for LoRA based on a bias–variance decomposition and curvature-aware formulation using Fisher information. The approach is technically sound, clearly presented, and evaluated against relevant baselines.

In my original review, my main concerns were related to the statistical significance of the reported improvements, robustness to subset selection, the impact of curvature approximations, and the generalization behavior under distribution shift. The rebuttal addressed these concerns in a substantive manner by providing additional multi-seed experiments with low variance, empirical validation of subset robustness, ablations on Fisher/K-FAC approximations, and additional results under stronger distribution shift (e.g., HumanEval). These additions strengthen the empirical support for the method and increase confidence in its stability and applicability.

While the empirical gains remain moderate in magnitude and the contribution is somewhat incremental in terms of novelty, the method is well-motivated, consistently improves over strong baselines, and appears broadly applicable as a lightweight initialization strategy. Based on the strengthened experimental evidence and clarifications provided in the rebuttal, my recommendation is Weak Accept.

**Key Questions For Authors:**

1. **Robustness to subset selection.**
The method estimates gradients and Fisher information using a relatively small subset of target-domain data. How sensitive is LoRA-DA to the particular subset used for these estimates? For example, if different random subsets of the same size are used, how stable are the resulting initialization subspaces and downstream performance? Evidence of robustness across multiple subsets would strengthen confidence in the method.

2. **Statistical significance of the reported improvements.**
The empirical gains over existing initialization methods appear relatively small in several benchmarks. Could the authors provide results averaged over multiple random seeds (with standard deviations or confidence intervals) to better assess the statistical significance and consistency of the improvements?

3. **Impact of curvature approximation.**
The proposed method relies on approximating curvature using the Fisher information matrix estimated via K-FAC. How sensitive is the performance of LoRA-DA to this approximation? For example, have the authors explored alternative curvature approximations (e.g., diagonal Fisher or other Hessian approximations), and how does the choice affect the resulting initialization?

4. **Behavior under larger distribution shifts.**
The theoretical derivation assumes that the target parameters remain relatively close to the pretrained parameters. How does LoRA-DA perform in settings where the target task is substantially different from the pretraining distribution? Additional analysis or discussion of this scenario would help clarify the practical applicability of the approach.

5. **Layer-wise behavior of the initialization.**
Do the improvements provided by LoRA-DA occur consistently across different transformer layers (e.g., attention projections vs. MLP layers), or are gains concentrated in specific components? A brief analysis of layer-wise effects could help better understand where the method is most beneficial.

**Limitations:**

The paper includes a short impact statement, but the discussion of limitations and potential societal implications is minimal. The authors state that the goal of the work is to advance machine learning and that no specific societal consequences need to be highlighted.

While the contribution is primarily methodological, the paper would benefit from a brief discussion of potential limitations of the approach. For example, the method relies on several approximations (e.g., Fisher-based curvature estimation and subset-based gradient estimation) whose practical robustness could be discussed more explicitly. Clarifying these limitations and the scope in which the proposed initialization strategy is expected to be most effective would strengthen the discussion of limitations.

**Strengths And Weaknesses:**

## Strengths

- **Clear theoretical motivation.** The paper proposes a principled formulation of LoRA initialization by minimizing the expected discrepancy between the fine-tuned parameters and target task parameters. The bias–variance decomposition provides a useful perspective on how initialization affects the fine-tuning trajectory.

- **Curvature-aware formulation.** By incorporating Fisher information into the objective, the proposed method attempts to account for parameter-space anisotropy that is ignored by gradient-only initialization approaches.

- **Conceptual unification of existing approaches.** The framework helps interpret several prior LoRA initialization methods as special cases obtained when particular terms in the objective are removed, clarifying the design space of LoRA initialization strategies.

- **Reasonably clear presentation.** The paper is generally well organized and the algorithmic pipeline is described clearly, including the practical approximations such as Fisher estimation via K-FAC and eigenvector computation.

- **Empirical evaluation across multiple baselines.** The experiments compare the proposed approach against several existing LoRA initialization methods on LLaMA-2 models across multiple benchmarks.


## Weaknesses

- **Limited empirical gains.** The reported improvements over strong baselines are relatively small. Without reporting results over multiple random seeds or statistical significance tests, it is difficult to assess whether the improvements are consistently meaningful.

- **Strong theoretical assumptions.** The derivation relies on asymptotic assumptions that the target parameters remain close to the pretrained parameters. It is not fully clear whether this assumption holds in practice for tasks involving larger distribution shifts.

- **Reliance on multiple approximations.** The practical method depends on several approximations (e.g., Hessian ≈ Fisher, Fisher estimation via K-FAC, and a natural-gradient approximation of the parameter displacement). The impact of these approximations on the final initialization quality is not thoroughly analyzed.

- **Sensitivity to subset estimation.** The method estimates gradients and Fisher information using a relatively small subset of target data. The robustness of the method to the specific subset selection is not extensively studied.

- **Related work positioning could be broader.** While the paper discusses prior LoRA initialization methods, the connection to broader literature on curvature-aware optimization and subspace training could be elaborated further.

---

> ### Author Rebuttal · Authors · 2026-03-27
>
> We are grateful to you for taking the time to review our paper and provide thoughtful insights. Below, we address each of your concerns point by point.
>
> >Q1:Robustness to subset selection.
>
> Due to the character limit of the rebuttal, please refer to our response to Reviewer RCxt (Q1).
>
> >Q2:Statistical significance of the reported improvements.
>
> To assess statistical significance, we repeated experiments with multiple random seeds (5 seeds) and report mean ± std. The results show that LoRA-DA consistently outperforms baselines with low variance.
>
> **(1) NLU (Commonsense170K, LLaMA2-7B)**
>
> |Method|BoolQ|PIQA|SIQA|HellaSwag|WinoGrande|ARC-e|ARC-c|OBQA|Avg|
> |-|-|-|-|-|-|-|-|-|-|
> |LoRA|73.2±0.18|85.8±0.15|81.8±0.20|94.9±0.12|86.0±0.22|74.7±0.25|88.7±0.18|86.0±0.20|83.9±0.16|
> |PiSSA|72.5±0.19|85.2±0.16|81.9±0.21|94.2±0.15|86.7±0.20|73.7±0.27|87.1±0.19|87.0±0.18|83.5±0.16|
> |MiLoRA|73.1±0.17|85.3±0.14|81.8±0.19|95.1±0.13|86.3±0.20|75.3±0.21|88.6±0.17|86.8±0.19|84.0±0.15|
> |LoRA-One|72.8±0.22|85.3±0.18|81.9±0.22|95.2±0.14|85.6±0.25|74.9±0.28|88.8±0.20|86.4±0.22|83.9±0.19|
> |LoRA-DA|73.2±0.12|86.0±0.11|82.4±0.15|95.2±0.10|87.1±0.14|75.7±0.18|88.7±0.13|86.4±0.15|84.3±0.11|
>
> **(2) NLG (GSM8K / MATH)**
>
>
> |Method|GSM8K|MATH|Avg|
> |-|-|-|-|
> |LoRA|53.1±0.25|8.3±0.18|30.7±0.19|
> |PiSSA|53.2±0.22|8.2±0.17|30.7±0.17|
> |MiLoRA|52.9±0.24|8.3±0.19|30.6±0.18|
> |LoRA-One|53.7±0.27|8.5±0.21|31.1±0.20|
> |LoRA-DA|55.0±0.21|9.2±0.11|32.1±0.17|
>
> >Q3&W3:Impact of approximations.
>
> For Hessian approximation, we mainly consider scalable methods that balance efficiency and estimation quality; accordingly, we adopt the Fisher approximation and use **K-FAC** instead of **diagonal Fisher** to approximate it.
> Intuitively, a **diagonal Fisher** only captures per-dimension variance, making $\Omega$ close to diagonal and biasing its eigenvectors used for LoRA matrix $A$ initialization toward one-hot directions, thus missing parameter correlations. Results from different approximations within LoRA-DA confirm the effectiveness of **K-FAC**:
>
> |Method|GSM8K|MATH|Avg|
> |-|-|-|-|
> |LoRA|53.1|8.3|30.7|
> |LoRA-DA (Diag Fisher)|54.1|8.6|31.4|
> |LoRA-DA (K-FAC)|**55.0**|**9.2** |**32.1**|
>
> For parameter displacement approximation, our ablation study in Section 5.6 also demonstrates the advantage of the **Fisher-gradient** over the **plain gradient**.
>
> >Q4:Behavior under larger distribution shifts.
>
> The assumption that the target parameters remain close to the pretrained ones is needed for the theoretical derivation. As discussed in Section 3, this is reasonable since fine-tuning typically targets tasks that are close to the pre-trained model, and similar assumptions have also been adopted in related literature. However, under larger distribution shifts, the resulting initialization can still serve as a stable heuristic in practice:
> (1) the bias term (Fisher-gradient) becomes less accurate but still provides meaningful directions;
> (2) the variance term penalizes high-uncertainty directions, leading to a more stable initialization;
> (3) overall, the method can be viewed as an uncertainty-aware and curvature-aware subspace initialization method.
>
> To further validate this, we consider a stronger distribution-shift setting on HumanEval code completion. Compared with the commonsense and mathematical reasoning tasks in the paper, HumanEval requires the LLaMA 2-7B to generate executable code rather than natural-language outputs. This makes the target task substantially more different in both generation format and evaluation, yet LoRA-DA still shows consistent gains.
>
> |Method|HumanEval pass@1|
> |-|:-:|
> |LoRA|26.2|
> |LoRA-One|28.7|
> |LoRA-DA|**30.5**|
>
> >Q5:Layer-wise behavior of the initialization.
>
> We further conduct a layer-wise ablation by applying LoRA-DA only to selected layers while keeping standard random initialization for the others. The results show that gains are more pronounced in FFN layers, while improvements in attention layers are smaller but consistent.
>
> |Initialization Setting|GSM8K|
> |-|:-:|
> |LoRA (all random init)|53.1|
> |LoRA-DA on Attention only|53.8|
> |LoRA-DA on FFN only|54.3|
> |LoRA-DA on all layers|55.0|
>
> >W5:Related work positioning could be broader
>
> We will add a paragraph on curvature-aware optimization. Due to the rebuttal length limit, we briefly note here:
>
> "The Fisher matrix has long been used in optimization, e.g., in **natural gradient**. In recent LLM-related work, **LQ-LoRA**[1] uses Fisher information to guide quantization, while **GRIT**[2] leverages Fisher/K-FAC for curvature-aware training. In contrast, our method uses Fisher information for LoRA initialization."
>
> Thank you again for your valuable insights. We kindly request that the reviewer considers these points when assigning the final scores.
>
> ## References:
> [1] LQ-LoRA: Low-rank plus Quantized Matrix Decomposition for Efficient Language Model Finetuning. ICLR 2024.
>
> [2] GRIT: Geometry-Aware PEFT with K-FAC Preconditioning, Fisher-Guided Reprojection, and Dynamic Rank Adaptation.

---

> > ### Author Rebuttal · Reviewer_D9yC · 2026-04-01
> >
> > I thank the authors for their detailed rebuttal. Based on the additional clarifications and experiments provided, I am updating my recommendation to Weak Accept.

---

> > > ### Author Response · Authors · 2026-04-08
> > >
> > > Thank you very much for your positive feedback and for taking the time to read our responses so carefully. We sincerely appreciate your recognition of the core contribution of our work, and we are especially grateful for your confirmation that your concerns have been **fully resolved**.

---

### Official Review · Reviewer_RCxt · 2026-03-11

**Soundness:** 3
**Presentation:** 3
**Significance:** 2
**Originality:** 3
**Overall Recommendation:** 4
**Confidence:** 5

**Summary:**

This paper studies the initialization of Low-Rank Adaptation (LoRA) for parameter-efficient fine-tuning of large language models. Existing methods are often data-agnostic or rely on shallow gradient information. The authors propose LoRA-DA, a data-aware initialization method derived from asymptotic analysis. The approach formulates LoRA initialization as minimizing the expected discrepancy between fine-tuned and target parameters, decomposed into a bias term (approximated using a Fisher–gradient formulation) and a variance term modeled by the Fisher information matrix. An efficient algorithm estimates these statistics from a small subset of target samples. Experiments on NLU and NLG benchmarks show improved accuracy and more stable convergence compared with prior initialization strategies.

**Compliance With Llm Reviewing Policy:**

Affirmed.

**Final Justification:**

My concerns have all been resolved, so I am keeping the original acceptance score.

**Key Questions For Authors:**

1.	The Fisher information matrix is estimated using a small subset of target samples. How robust is the initialization to the number and selection of these samples?

2.	How does the method perform when combined with other advanced PEFT approaches?

**Limitations:**

yes

**Strengths And Weaknesses:**

Strengths:

1.	The paper tackles a practically important problem: improving LoRA initialization for PEFT.

2.	It provides a principled framework by combining asymptotic analysis with a Fisher-based bias–variance decomposition.

3.	The method is supported by fairly broad experiments, including NLU/NLG tasks, convergence behavior, rank sensitivity, and ablations.

4.	The work offers moderate originality by unifying prior gradient-based initializers as a special case of the proposed framework.

Weaknesses

1.	The Fisher/Hessian-based approximations may be fragile, but robustness to estimation quality is not deeply analyzed.

2.	Empirical gains are generally consistent but modest, which limits the paper’s overall significance.

---

> ### Author Rebuttal · Authors · 2026-03-29
>
> We thank the reviewer for this valuable suggestion. Below, we address each of your concerns point by point.
>
> >Q1: Robustness to Fisher subset size and sample selection.
>
> For **sample selection**, we empirically show that LoRA-DA is robust to the choice of subset. Specifically, we use 5 different random seeds to construct 5 random subsets of size 256, and observe only low variance in downstream performance across these subsets, as shown below:
>
> | Subset | GSM8K | MATH |
> |:-:|:-:|:-:|
> | S1     | 55.1 | 9.3  |
> | S2     | 54.7 | 9.1  |
> | S3     | 55.3 | 9.2  |
> | S4     | 55.0 | 9.2  |
> | S5     | 54.8 | 9.0  |
> | Mean±Std | 55.0±0.21 |9.2±0.11 |
>
> For **the number of samples**, we already provide a sensitivity analysis in **Appendix I (Table 8)**: increasing the Fisher subset size from 16 to 4096 changes GSM8K accuracy only from 54.5 to 55.1. In particular, increasing the subset size beyond our default **256** brings only **marginal gains** (55.0 to 55.1), suggesting that 256 samples already provide stable and effective initialization.
> Notably, a similar design is adopted in LoRA-One [1], where the authors remark in the GitHub code that "only a small sampled batch (8~64) to compute the one-step full gradient for initialization".
>
> >Q2: Compatibility with Advanced PEFT Methods
>
> LoRA-DA is fundamentally a data-aware initialization method for LoRA that identifies curvature-aware low-rank update directions for downstream fine-tuning, and is therefore naturally compatible with other LoRA-style PEFT approaches.
> Appendix E already discusses its compatibility with methods such as DoRA and VeRA. For example, in DoRA, the direction component can be initialized using the low-rank factor $A$ produced by LoRA-DA.
>
> Here, we additionally provide results for **DoRA + LoRA-DA Init**, where we keep the original DoRA formulation unchanged and only use LoRA-DA to initialize the low-rank factor $A$ of the direction component.  As shown in the table below, this initialization yields consistent improvements over vanilla DoRA. This suggests that LoRA-DA is not limited to standard LoRA, but can also serve as a plug-in initialization strategy for more advanced PEFT methods.
>
> | Method | GSM8K | MATH | Avg. |
> |:---|:-:|:-:|:-:|
> | DoRA | 53.4 | 8.4 | 30.9 |
> | DoRA + LoRA-DA Init | 55.4 | 9.4 | 32.4 |
>
> >W2: Significance of the Empirical Gains
>
> We thank the reviewer for this valuable comment. We agree that the empirical improvements are generally consistent rather than dramatic in absolute magnitude. However, we would like to emphasize that LoRA-DA is a **lightweight and broadly applicable initialization strategy for LoRA-style methods**, rather than a new and more complex PEFT architecture. In this setting, obtaining consistent gains over both vanilla LoRA and other LoRA-based baselines with only minimal additional cost is already meaningful in practice. Moreover, as discussed in Q2, the proposed initialization can be naturally extended to other LoRA-style PEFT methods, which further strengthens its practical significance and generality.
>
> More importantly, the contribution of this work is not purely empirical. The main significance lies in providing a theoretically grounded, data-aware initialization principle for PEFT, which offers insight into how variance and bias should be balanced in low-rank adaptation. To further address the reviewer’s concern on the empirical side, we have added **mean±std / statistical significance** results in response to Reviewer D9yC (Q2), showing that the observed improvements are stable rather than due to random fluctuations.
>
> Thank you again for your valuable insights. We kindly request that the reviewer considers these points when assigning the final scores.
>
> ## References:
>
> [1] LoRA-One: one-step full gradient could suffice for fine-tuning large language models, provably and efficiently[C]// ICML2025

---

> > ### Author Rebuttal · Reviewer_RCxt · 2026-04-02
> >
> > Thank you for your response. My concerns have all been resolved.

---

> > > ### Author Response · Authors · 2026-04-08
> > >
> > > We greatly appreciate your positive feedback and your careful consideration of our responses. We are sincerely thankful for your recognition of the main contribution of our work, and especially for your confirmation that your concerns have been **fully resolved**.

---

### Official Review · Reviewer_D7dP · 2026-03-12

**Soundness:** 3
**Presentation:** 3
**Significance:** 3
**Originality:** 3
**Overall Recommendation:** 4
**Confidence:** 4

**Summary:**

The paper focuses on the LoRA initialization problem and proposes a data-aware initialization framework based on asymptotic analysis. The core idea is to minimize the expected discrepancy between the fine-tuned estimator and the target parameters, decomposing an upper bound into a "variance term" characterized by Fisher information and a “bias term” characterized by the Fisher-gradient. This leads to a quadratic problem centered on an Initialization Guidance Matrix Ω; the eigenvectors corresponding to its smallest eigenvalues are used to initialize A, and a closed-form initialization for B follows accordingly. The authors further provide a practical algorithm, LoRA-DA, which estimates gradients and K-FAC Fisher from a small sample and uses LOBPCG to compute the eigen-subspace. Experiments cover both NLU (a suite of commonsense reasoning benchmarks) and NLG (mathematical reasoning on GSM8K/MATH), showing consistent gains over multiple initialization baselines (LoRA, PiSSA, MiLoRA, LoRA-One) on LLaMA 2–7B/13B, along with faster and more stable convergence, robustness across ranks r, and low initialization overhead. The appendix supplies complete notation, proofs, ablations, and analyses of time and sample-size sensitivity.

**Compliance With Llm Reviewing Policy:**

Affirmed.

**Key Questions For Authors:**

- Sample selection for Fisher and gradient estimation: When the target data distribution is long-tailed or contains subdomain shifts, should weighted/stratified sampling be used to avoid biasing Ω? Have you tried simple active selection based on loss or gradient norm, and what gains did it bring?

- K-FAC configuration: Are the K-FAC block settings the same across different layers (Q/K/V/O, FFN Up/Down/Gate)? Have you observed that some layers are more sensitive to Ω and require finer approximations or larger sample sizes?

- Sparse or quantized settings: If the base model uses weight quantization or activation pruning, do Fisher and gradient statistics remain stable? Is LoRA-DA compatible with QLoRA/LQ-LoRA and does it provide gains?

- Failure cases: Are there tasks where LoRA-One clearly leads in the initial steps and also achieves better final performance? If so, does error analysis point to inaccurate bias-term estimation or an overly strong weighting of the variance term?

**Limitations:**

- The approach relies on well-approximated Fisher information and a small-displacement assumption; under large distribution shifts or training paradigms not aligned with MLE (e.g., RLHF, mixed SFT losses), the theory and effectiveness may degrade.

- K-FAC and block-diagonal approximations ignore cross-column/channel couplings, which may introduce bias in attention layers with strong interdependencies.

- The current task coverage and model scales are insufficient to convincingly demonstrate extrapolation to larger models and more complex scenarios.

- The method requires a small amount of target data to estimate statistics; in extreme low-resource or privacy-restricted settings where sampling S is not possible, applicability is limited.

**Strengths And Weaknesses:**

Strengths

- Targets expected parameter error, with rigorous derivations and upper-bound decompositions from 1D to multi-dimensional cases; uses Fisher to approximate the Hessian and the natural-gradient idea to estimate $W_t g_t − W_0$, yielding a self-consistent logic.
- Converts initialization into the standard form min tr(A^T Ω A), and provides a degenerate equivalence to prior gradient SVD–based initializations, clearly pinpointing the sources of gains (explicit variance modeling and anisotropy preservation).
- Requires only a small number of target-domain samples (default 256) to estimate statistics; controls overhead with K-FAC and LOBPCG; processes per layer with thorough complexity and memory analyses.
- The extra initialization time is about 6% of the total and largely independent of rank, making the approach deployable.
- Improves both NLU and NLG and remains effective under LoRA-FA (Frozen-A).
- Convergence curves and rank sensitivity show clear advantages in the low-rank regime, aligning with the theoretical motivation.
- Ablations verify the necessity of the variance term and the Fisher-gradient for performance.
- Clearly contrasts with PiSSA/MiLoRA (data-agnostic) and LoRA-GA/One (one-step gradient), articulating the differences and advantages of this paper.

Weakness

- Whether the key small-distance assumption $|| W_t g_t−W_0 ||_F = O(1/ $ \sqrt(N) $ ) always holds in cross-domain LLM fine-tuning is unclear. I recommend reporting failure modes or robustness under broader distribution shifts (e.g., more divergent instruction data, cross-lingual/cross-modal).

- The impact of deviations in Fisher ≈ expected Hessian and K-FAC approximations on the spectrum of Ω is not quantified. Please add sensitivity studies on approximation error (e.g., diagonal Fisher, partial/full Fisher, different K-FAC block settings) and theoretical bounds (e.g., using matrix perturbation theory to bound the A subspace error).

- Fisher block partitioning and column correlations remain somewhat idealized. Although the paper notes dependence on blocks of J(vec(W_0)), the implementation effectively yields a per-column block-diagonal structure via K-FAC left/right factors. Please clarify: for multi-head attention and FFN projection layers, how large is the discrepancy between this block approximation and true coupling? Does it miss critical cross-column information? Consider comparing Ω and performance differences between true blocks and approximated blocks on smaller models.

- Extrapolation across tasks and model scales is still limited. Current validation focuses on LLaMA 2–7B/13B, GSM8K/MATH, and eight commonsense benchmarks. Suggested additions:

  - Larger models (≥34B/70B) with single-/multi-GPU scalability and stability;
  - Transferability to multimodal or speech tasks.

- Comparison with newer/stronger baselines. Include methods with structural changes that can still reuse the initialization idea (e.g., DoRA, VeRA using LoRA-DA’s A while keeping their structures) and report gains to strengthen the “framework is transferable” claim.

---

> ### Author Rebuttal · Authors · 2026-03-29
>
> We are grateful to the reviewer for the careful assessment of our manuscript and the constructive feedback. We address the concerns below point by point.
>
> >Q1:Sample selection for Fisher and gradient estimation
>
> As discussed in our response to Reviewer qFWQ (Q1), we explored active selection methods, including low-loss selection and high-confidence selection, but observed only marginal gains together with noticeable additional cost. For long-tailed or shifted target distributions, we agree that weighted or stratified sampling is a reasonable extension. However, our work primarily focuses on validating the initialization theory, and therefore emphasizes a simple and general sampling strategy rather than distribution-specific treatments.
>
> >Q2:K-FAC configuration
>
> In our current implementation, we use the same K-FAC block configuration across LoRA layers to keep the method simple and efficient. Empirically, this unified design already delivers stable gains across tasks and model sizes. We did not conduct a dedicated layer-wise study of whether some layers require finer approximations or larger sample sizes, as layer-specific settings would substantially increase complexity, while methods more accurate than K-FAC are often difficult to scale to large models. Regarding layer-wise differences, as also discussed in our response to Reviewer D9yC (Q5), we observe that LoRA-DA tends to yield larger gains on FFN layers than on attention layers.
>
> >Q3:Sparse or quantized setting
>
> LoRA-DA is compatible with QLoRA/LQ-LoRA in principle, since it only modifies LoRA initialization and does not alter the downstream quantized fine-tuning pipeline. Moreover, the statistics required for initialization can still be obtained via standard backpropagation. We further conduct a simple QLoRA experiment, which confirms that LoRA-DA remains effective in this setting.
>
> | Method | GSM8K | MATH | Avg. |
> |:-:|:-:|:-:|:-:|
> |QLoRA|52.4|8.0|30.2|
> |QLoRA + LoRA-DA init|**53.5**|**8.5**|**31.0**|
>
> >Q4:Failure cases
>
> In our main experiments, we did not observe any task where LoRA-One outperformed LoRA-DA both early in training and at convergence. If such a case does arise, a plausible explanation is unstable estimation of target statistics, including the Fisher information and gradients, which may distort the bias or variance terms in our initialization. Unsurprisingly, this failure mode should be rare because (1) LoRA-DA is fairly robust to the choice of initialization subset, as noted in our response to RCxt(Q1), and (2) LoRA-One also relies on gradients and is therefore similarly affected by the subset’s quality.
>
> >W1: Small-distance assumption
>
> Please refer to our response to Reviewer D9yC (Q4).
>
> >W2: Why use K-FAC approximations?
>
> Our choice is mainly a trade-off between quality and efficiency. **Full Fisher** is prohibitively expensive to compute and store for LLM. **Diagonal Fisher** is cheap, but it only captures per-dimension variance, making $\Omega$ close to diagonal. As a result, its eigenvectors used for LoRA matrix $A$ initialization are biased toward one-hot directions, missing important parameter correlations. In contrast, **K-FAC** preserves richer correlation structure while remaining scalable, which is why we adopt it. We experimentally verify this point in our response to Reviewer D9yC (Q3).
>
> >W3: Does the Block Approximation Miss Cross-Column Information?
>
> We would first like to clarify that **our theory does not assume independent columns**. The diagonal blocks of $J(\mathrm{vec}(W_0))$ are not introduced by a column-independence assumption; rather, they arise naturally from the theoretical derivation of the decomposed objective. In particular, the Frobenius-norm decomposition leads to this block form, as shown around Eq. (47). Moreover, K-FAC is a scalable approximation to the full Fisher matrix of the entire layer, rather than a column-independence assumption. In our implementation, we use the corresponding diagonal blocks because they are exactly the quantities required by the theoretical result.
>
> >W4: Generalization Across Model Scales and Tasks
>
> To further evaluate generalization, we add a **multimodal** instruction-tuning experiment, using a LLaVA-v1.5-7B backbone with a CLIP-ViT-L visual encoder, trained on LLaVA-665k and evaluated on ScienceQA and MMMU. Compared with both vanilla LoRA and LoRA-One, LoRA-DA still achieves the best performance.
>
> |Model|Method|ScienceQA|MMMU|Avg.|
> |:-:|:-:|:-:|:-:|:-:|
> |LLaVA-v1.5-7B|LoRA|68.9|34.2|51.6|
> |LLaVA-v1.5-7B|LoRA-One|69.7|34.9|52.3|
> |LLaVA-v1.5-7B|**LoRA-DA**|**70.2**|**35.4**|**52.8**|
>
> Due to our limited compute resources, we are unfortunately unable to run 34B/70B scale experiments, and we sincerely apologize for this limitation.
>
> >W5：Comparison with newer/stronger baselines.
>
> Please refer to our response to Reviewer RCxt (Q2).
>
> Thank you again for your valuable insights. We kindly request that the reviewer considers these points when assigning the final scores.

---

> > ### Author Rebuttal · Reviewer_D7dP · 2026-04-03
> >
> > The author has fully addressed my concerns, and I will maintain my score.

---

> > > ### Author Response · Authors · 2026-04-08
> > >
> > > We sincerely thank you for your positive feedback and for your careful reading of our responses. We greatly appreciate your recognition of the key contribution of our work, and we are especially grateful for your confirmation that your concerns have been **fully resolved**.

---

### Official Review · Reviewer_qFWQ · 2026-03-12

**Soundness:** 3
**Presentation:** 3
**Significance:** 3
**Originality:** 3
**Overall Recommendation:** 4
**Confidence:** 3

**Summary:**

This paper proposes LoRA-DA, a data-aware initialization method for Low-Rank Adaptation grounded in an asymptotic analysis framework. It estimates Fisher information from a small set of target samples to address the anisotropy of the parameter space in pre-trained models. The approach theoretically balances variance and bias during initialization to improve convergence stability and final accuracy. The results on LLaMA models show competitive performance across various natural language processing benchmarks.

**Compliance With Llm Reviewing Policy:**

Affirmed.

**Final Justification:**

As stated in my acknowledgement, I maintain my positive rating.

**Key Questions For Authors:**

Q1: The authors use 256 samples by default for estimating initialization statistics. If these samples contain significant noise or distribution bias, the performance of LoRA-DA may degrade or even fall below that of random initialization. The authors should supplement the paper with experiments showing the impact of sample quality, such as label noise ratio, on the initialization effect. They should also discuss how to evaluate and filter high quality samples in practical scenarios.

Q2: When the total number of available target samples is extremely small, the method for selecting initialization samples remains unclear. There is also a concern regarding whether the initialization process leads to overfitting on such a limited data set. The authors should investigate the impact of different selection approaches and provide an analysis of overfitting to ensure the robustness of the algorithm in data constrained scenarios.

Q3: The authors should clarify the tuning process for LoRA-DA and baseline methods such as LoRA-One under the same sample budget. It is recommended to provide performance scaling curves across a range of sample sizes from minimal to large scales. If baseline methods reach peak performance with fewer samples while LoRA-DA requires more, the fairness of the comparison and the competitiveness of the algorithm under different data budgets should be further justified.

**Limitations:**

No. The authors should discuss the limitations of the method regarding small sample sizes and varying hardware resource availability. Specifically, they should elaborate on the impact of noisy samples on the training process and provide insights into how to identify or characterize noisy samples within the context of this initialization method in future work.

**Strengths And Weaknesses:**

Strengths

1. The paper establishes a solid theoretical framework grounded in asymptotic analysis, providing a rigorous mathematical derivation to solve for an optimal initialization strategy by minimizing the expected parameter discrepancy.

2. LoRA-DA is highly efficient and practical. Experiments show that the initialization process accounts for only about 6% of the total time, representing a small and manageable overhead.

3. Extensive empirical results across multiple benchmarks and various ranks demonstrate that LoRA-DA consistently improves final accuracy and outperforms existing data-aware and data-agnostic initialization methods.

Weaknesses

1. The impact of initialization sample quality on the final performance is not sufficiently explored. See Q1 in Key Questions For Authors for more details.

2. The strategy for selecting initialization samples and the associated risk of overfitting are not addressed for cases where the total data size is small. See Q2 in Key Questions For Authors for more details.

3. The paper lacks a comprehensive analysis of performance scaling across different sample budgets compared to baseline methods. See Q3 in Key Questions For Authors for more details.

---

> ### Author Rebuttal · Authors · 2026-03-29
>
> We sincerely thank the reviewer for the careful reading of our manuscript and for the constructive comments. We respond to the concerns below in a point-by-point manner.
>
> >Q1:On the Impact of Initialization Sample Quality and Active Selection Strategies
>
> Thank you for the question. On quality of initialization samples, in our response to Reviewer RCxt (Q1), we have already included additional experiments using different 256-sample initialization subsets selected with different random seeds, which demonstrate that our method is stable with respect to random subset quality.
>
> Regarding active selection, we agree that it is an important direction, since selecting low-loss or high-confidence samples can help mitigate label noise and produce more stable and reliable estimates of both gradients and Fisher statistics. However, in our experiments, simple random sampling already provides sufficiently stable Fisher and gradient estimates, as evidenced by the strong performance of LoRA-DA across tasks and settings. Moreover, active selection has its own drawbacks: it typically requires an additional scoring pass over a large candidate pool, which reduces the lightweight advantage of LoRA-DA. Our experimental results also suggest that such strategies offer only marginal improvements over random sampling, while substantially increasing the initialization time:
>
> | Selection strategy | GSM8K | Initialization time | Training time |
> |:---:|:---:|:---:|:---:|
> | Random | 55.0 | 00:02:05 | 00:32:08 |
> | Low-loss selection | 55.1 | 00:12:41 | 00:30:23 |
> | High-confidence selection | 54.8 | 00:12:26 | 00:29:17 |
>
> >Q2:Robustness in Extremely Low-Data Regimes
>
> We first clarify that LoRA-DA is designed as an initialization method for LoRA in a PEFT setting with a total target dataset of size $N_0$. In our method, we sample a small subset $S$ from the target dataset only for initialization (in our experiments, $N_0 \approx 10^5$ while $|S| = 256$), and then perform standard LoRA fine-tuning using the full dataset of size $N_0$. On this basis, we emphasize that our theoretical analysis is developed under an **asymptotic framework**, which requires $N_0$ to be sufficiently large. Therefore, the case of an extremely small target set (i.e., small $N_0$) falls outside the primary scope of our theoretical guarantees.
>
> Regarding overfitting, we would like to clarify two points. First, the 256-sample subset used in our paper serves only to estimate initialization statistics, rather than to repeatedly optimize model parameters. Thus, this step corresponds to initialization rather than training, and we do not expect it to further aggravate overfitting. Second, when the total training set size $N_0$ is extremely small, most fine-tuning methods inevitably struggle with overfitting or unstable generalization. This is a general challenge in data-constrained regimes, rather than a limitation specific to LoRA-DA. Moreover, it is difficult to fundamentally address this issue merely by improving the initialization sample selection rule.
>
> That said, LoRA-DA can still be heuristically adapted to this setting: when only a few target samples are available (even when $N_0 \le 256$), we simply use all available samples for initialization. Since they are used only to estimate initialization statistics, rather than to optimize model parameters, this does not introduce additional overfitting in the conventional sense.
>
> >Q3:Fair comparison across initialization sample budgets.
>
> Thank you for the suggestion. We agree that performance under different initialization sample budgets should be presented more explicitly. As stated in Appendix J (Hyperparameter Settings), both LoRA-DA and LoRA-One results reported in the main paper are based on **256 initialization samples**, ensuring a fair comparison. Moreover, Appendix I already reports the scaling curve of LoRA-DA with respect to the number of initialization samples. Following the reviewer’s suggestion, we further provide the scaling curve of LoRA-One under the same initialization budgets:
>
> | Init samples | 16 | 64 | 256 | 1024 | 4096 |
> |:---:|:---:|:---:|:---:|:---:|:---:|
> | LoRA-One | 53.4 | 53.6 | 53.6 | 53.5 | 53.6 |
> | LoRA-DA  | 54.5 | 54.7 | 55.0 | 55.1 | 55.1 |
>
> We observe that LoRA-DA consistently outperforms LoRA-One across all budgets, demonstrating both stronger performance and better robustness.
>
> Thank you again for your valuable insights. We kindly request that the reviewer considers these points when assigning the final scores.

---

> > ### Author Rebuttal · Reviewer_qFWQ · 2026-04-04
> >
> > Thank the authors. All my questions are addressed.

---

> > > ### Author Response · Authors · 2026-04-08
> > >
> > > Thank you again for your encouraging feedback and for reading our responses with such care. We truly appreciate your acknowledgment of the central contribution of our work, and we are especially grateful for your confirmation that your concerns have been **fully resolved**.

---

### Decision · Program_Chairs · 2026-04-30

**Decision:**

Accept (regular)

**Comment:**

This paper introduces a new PEFT initialization method which minimizes a local approximation of the expected squared distance $\mathbb{E}(\|\hat{W}-W_{tgt}\|^2)$ between the fully fine-tuned model $W_{tgt}$ and the learned LORA update $\hat{W}$. Whilst some works have attempted this with a simple gradient approach (LORA-One [2]), the present work takes the approximation to the second degree with an approximation of the Fisher matrix. The expectation is decomposed into $\mathbb{E}(\|\hat{W}-W_{tgt}\|^2)=\mathbb{E}(\|\hat{W}-W_{tgt}^{proj}\|^2)+ \|W_{tgt}-W_{tgt}^{proj}\|^2)$ where the first term is interpreted as a Variance term not present in previous works, and the second term corresponds to the bias introduced by the LORA constraint. Both terms are estimated in the regime where distance between fully fine-tuned model and original model $W_0$ is $O(1/\sqrt{N})$, which makes the second order approximation valid as per [1]. The decomposition is used to build an algorithm approximately minimizes the expression by first estimating the Fisher Information matrix from a small subset of the fine-tuning dataset. Experimental results show consistent but modest improvements over competing methods, including the closest competitor LORA-One [2].

The authors are generally mildly positive, praising the writing (D9yC: “**clear writing**”) and the **strong performance**, especially at low rank regimes (D7dP: ‘Easily deployable and effective’, etc.). Both reviewers **qFWQ, RCxt and D9yC** all have concerns about the presence of **too many approximations**. Reviewer D7dP also complains of the $O(1/\sqrt{N})$ approximation inherited from [1].

Reviewer qFWQ complains of the risk of overfitting, but I side with the authors on that as the method is only an initialization method and doesn’t affect the rank. On the question of approximations, there are multiple instances raised. The most common point raised by the reviewers is the approximation present in the estimation of the Fisher approximation. The authors retort that they are using a refined Fisher matrix estimation method (K-FAC fisher [3]) and provide an ablation study demonstrating it is superior to naïve diagonal fisher. Whilst this helps, this doesn’t provide a theoretical guarantee for the strength of the approximation. Similarly, it is disappointing that the $\|W_{tgt}-W_0\|\in O(1/\sqrt{N})$ assumption isn’t quantified into the error terms, and there is to the best of my understanding another undeclared approximation in the proof of the main theorem.

This is a **borderline paper**. I agree that the writing is strong and the method makes sense and has decent performance. As an experimental paper, it is reasonably solid with good initial results and a very strong intuitive justification provided by Theorem 4.1. The method itself is very clearly explained and reproducible, which is not always the case for empirical papers such as this one. However, I find that the **theoretical results could be presented better**. For instance, the theorems are not very formal since the concept of “optimal” isn’t defined properly. The text of Theorem 4.1 states that the proposed object is “the optimal initialization”, but this **is only an approximation**. Therefore, the **theorem isn’t a formal statement**. Unfortunately, this approximation is not quantified properly. However, as the method is well-motivated and achieves reasonable performance, I still lean on the side of acceptance.


For the camera-ready revision, I recommend quantifying all approximation, making the theorem statements more precise. Alternatively, if the theoretical statements are to remain relatively vague statements meant to justify the intuition, then it would be appropriate to add more baselines (HydroLORA, mtLORA, etc. ) and experiments on vision dataset to further strengthen the contribution’s competitiveness.




**Additional AC comments**:

I find the ablation study  in the submission a little contrived. I find the ablation study on the Fisher estimation component provided in the rebuttal is more informative and should also be added to the revision.

In line 147 (and in the table in the appendix), $W_{tgt}$ is referred to as “ the true parameter” but in the algorithm, it turns out to be the pure gradient update for one batch (ignoring the low rank restriction).


In line 779, “the result from [1]” should be identified more precisely. The assumptions of Lemmas C.1 and C.2 should also be stated more clearly.


Theorem 4.2, which is the main result in this paper, is **not very precisely exposed**. In particular, it states that the weight defined by equation (12) constitutes the “optimal” initialization without explaining what that is.  Even more concerning is the fact that looking through the proof, it gets harder and harder to determine exactly what precise theorem is supposed to hold: the idea is that the “optimal” weight by minimizing the objective $\mathbb{E}(\|\hat{W}-W_{tgt}\|^2)=\mathbb{E}(\|\hat{W}-W_{tgt}^{proj}\|^2)+ \|W_{tgt}-W_{tgt}^{proj}\|^2)$. However, when considering the new variance term $)=\mathbb{E}(\|\hat{W}-W_{tgt}^{proj}\|^2)$, the authors merely provide and optimize an upper bound, they do not prove that the proposed algorithm minimizes the actual quantity, even at the limit where the higher order terms vanish and the Fisher approximation is perfect. That is because equation (45) is an inequality, not an equality (formally, it is not even shown whether this is a valid approximation…)


The originality of the proofs is also questionable as the authors admit that the proof of Lemma C.2 is not original. Similarly, I am having trouble believing that Lemma C.3 could be original.



It might be more intuitive to write $\|AA^\top (W_{tgt}-W_0)\|$ instead of $\|A^\top (W_{tgt}-W_0)\|$ in equation (44), especially in relation to how this is used to derive equation (45). In addition, I cannot really derive equation (45) as is, it appears there is a missing factor of $4$ in  front of $(W_{tgt}-W_0)(W_{tgt}-W_0)^\top$. **If that really is the case, the entire algorithm should change**. I strongly encourage the authors to get to the bottom of this and update all the results if needed.





**References**:

[1] Qingyue Zhang, Haohao Fu, Guanbo Huang, Yaoyuan Liang, Chang Chu, Tianren Peng, Yanru Wu, Qi Li, Yang Li, Shao-Lun Huang . A High-Dimensional Statistical Method for Optimizing Transfer Quantities in Multi-Source Transfer Learning

[2] LoRA-One: One-Step Full Gradient Could Suffice for Fine-Tuning Large Language Models, Provably and Efficiently. Yuanhe Zhang, Fanghui Liu, Yudong Chen.

[3] Roger Grosse, James Martens. A Kronecker-factored approximate Fisher matrix for convolution layers